# Resolving optimal ionomer interaction in fuel cell electrodes via operando X-ray absorption spectroscopy

Mengnan Wang[1,2], Jiaguang Zhang[3], Silvia Favero[1], Luke J. R. Higgins[4], Hui Luo[1], Ifan E. L. Stephens [2] ✉ & Maria-Magdalena Titirici [1,5] ✉

To bridge the gap between oxygen reduction electrocatalysts development and their implementation in real proton exchange membrane fuel cell electrodes, an important aspect to be understood is the interaction between the carbon support, the active sites, and the proton conductive ionomer as it greatly affects the local transportations to the catalyst surface. Here we show that three Pt/C catalysts, synthesized using the polyol method with different carbon supports (low surface area Vulcan, high surface area Ketjenblack, and biomass-derived highly ordered mesoporous carbon), revealed significant variations in ionomer-catalyst interactions. The Pt/C catalysts supported on ordered mesoporous carbon derived from biomass showed the best performance under the gas diffusion electrode configuration. Through a unique approach of operando X-ray Absorption Spectroscopy combined with gas sorption analysis, we were able to demonstrate the beneficial effect of mesopore presence for optimal ionomer-catalyst interaction at both molecular and structural level.

With the global urge of carbon neutrality, it becomes more and more critical to switch to more sustainable energy sources with zero carbon emissions, such as hydrogen. Proton Exchange Membrane Fuel Cells (PEMFCs) sit at the core of the $H_2$ economy allowing the conversion of $H_2$ fuel into clean energy with water as the only by-product[1]. However, the extensive application of PEMFC is restricted by the limited supply and high cost of commercial Pt-based catalysts for the cathodic oxygen reduction reaction (ORR). Despite the rapid and ever-growing progress in non-precious metals, commercial PEMFCs solely employ Pt based catalysts due to their superior activity and stability in acidic environment[2,3]. For context, the US Department of Energy (DOE) target for 2020 specifies that the total Platinum Group Metal content (across both electrodes) should be <0.125 g/kW (rated, gross) @ 150 kPa (abs)[4], whereas the Toyota Mirai currently on the market is estimated to contain contains around 30 g of platinum, equating to 0.26 g PGM

per kW[5]. Furthermore, even with the progress in non-PGM based electrocatalyst development for PEMFCs, led by Fe-N-C, this alternative still suffers from higher overall costs, unassessed environmental impacts, and the need for a 200% performance improvement to meet Pt-based PEMFC efficiency targets, as indicated by a comprehensive LCA study[6]. Therefore, there is a need to improve catalyst activity or utilization and ultimately reducing Pt loadings[7,8].

Over the recent decades, many research outputs have been published with the aim to improve the oxygen reduction catalytic activity, including tailoring the size of Pt nanoparticle/nanoclusters[9,10], alloying Pt with other metals such as Ni[8], Co[11], Pd[12] and developing unique nanostructures including nanowires[13], nano frameworks[14], core-shell[15] etc. Nevertheless, obstacles were encountered when translating those high mass activity electrocatalysts into efficient catalyst layers in PEMFC. This is because most of these electrocatalysts have been tested

[1]Department of Chemical Engineering, Imperial College London, London SW7 2AZ, UK. [2]Department of Materials, Royal School of Mines, Imperial College London, London SW7 2BP, UK. [3]Department of Chemistry, University of Lincoln, Lincoln LN6 7TS, UK. [4]Diamond Light Source, Harwell Science & Innovation Campus, Didcot OX11 0DE, UK. [5]Advanced Institute for Materials Research (AIMR), Tohoku University, 2-1-1 Katahira, Aoba-ku, Sendai 980-8577, Japan. ✉e-mail: i.stephens@imperial.ac.uk; m.titirici@imperial.ac.uk

in a rotating electrode disk/ring configuration where the current is limited by the low solubility of $O_2$ in the electrolyte, causing significant differences in the catalytic performance compared to a membrane electrode assembly (MEA) in real PEMFCs, where the operation current is 100-fold higher and the transportation of $O_2$, protons, electrons and water all play a critical role. However, the high amount of catalyst needed, and the time required for MEA measurements hindered its application in the design and optimization of new catalysts. Recent advancements in gas diffusion electrode (GDE) half-cell setups have emerged as a promising tool for examining the influence of catalyst layer properties on ORR performance[16–19]. According to an inter-lab comparison work, when utilizing consistent electrochemical protocols and ensuring homogeneous catalyst layers, tests with Pt-based GDEs have demonstrated consistent and comparable results across different setups and labs, offering insights more reflective of actual device conditions[20]. This paves the way for a more realistic representation of fuel cell conditions and a deeper understanding of catalyst stability in real-world electrochemical energy conversion devices.

Proton conductive ionomers such as Nafion™, which is a perfluorinated sulfonic acid (PFSA) ionomer, are often added to the mixture of catalysts and support, helping the catalyst layer binding together and proton conduction. However, an important but often overlooked aspect of this configuration, is the interaction between the carbon support, the proton conductive ionomer and the catalyst, which greatly affects the local transport of oxygen, protons, and water to the catalyst surface. The ionomer presence on the catalyst's surface, while improving proton transport and mechanical stability, can also have a poisoning effect[21] and add an extra diffusion layer for $O_2$ transport[22]. Nanostructured thin-film (NSTF) catalysts, developed by 3 M, have been the most extensively studied ionomer-free electrodes in recent decades[23]. NSTF catalysts use highly oriented, crystalline organic whiskers as a support structure, onto which thin films of Pt or Pt alloys are sputter-coated[24]. This unique thin film structure enables direct contact between the catalyst and the electrolyte, thus eliminating the need for an ionomer to facilitate proton transport. However, this thin structure also suffers from limited water-handling capacity[25]. The rate of water formation at the NSTF cathode is much higher than that of conventional Pt/C-ionomer electrodes, easily leading to flooding. Moreover, Pt/NSTF catalysts perform poorly under dry conditions due to reduced proton conductivity, making water management a critical challenge. Despite several material modification strategies aimed at addressing these issues[26], the inherent difficulties of eliminating ionomer from the catalyst layer remain.

Various strategies have been employed to manipulate the ionomer's coverage on Pt surface including the ionomer's chemistry modification[27], molecular masking[28], regulating ionomer distribution by introducing N sites[29] and catalyst support engineering[30]. Researchers have long recognised the effect of carbon support nanostructure to ionomer distribution and proposed a balance should be achieved between -low surface area carbon and high surface area carbon[31]. For example, Yarlagadda et al. examined 11 different commercial Pt/C catalysts consisting of Pt nanoparticles deposited on various carbon supports. They set out to test a hypothesis by measuring proton accessibility under dry conditions and evaluating local oxygen accessibility. Their findings indicated that low surface carbon supports experience direct contact between the ionomer and Pt, leading to local $O_2$ resistance. Conversely, high surface carbon supports showed limited accessibility to protons and reactant oxygen[30]. A pivotal conclusion from their research was the assertion that mesopores between 4 and 7 nm are the most accessible, and thus, could enhance the ORR performance. This conclusion was drawn based on the outlined 4–7 nm pore volume in their statistical analysis of the carbon support structure. While their work is foundational and offers intuitive insights, it's essential to note that the nanostructure of catalyst supports is intricate. The emphasis on the 4-7 nm pore volume,

though significant, might not capture the entire picture. While their study has been influential in identifying key relationships between carbon support nanostructure and ionomer distribution, it is essential to address certain shortcomings in their experimental methodology and absence of essential characterisations, which could undermine the robustness of their conclusions. The complexity of the catalyst support's nanostructure suggests that a more uniform porous structure needs to be developed to make a definitive claim about the impact of the mesoporous structure. The differences highlighted in their paper could potentially correlate with other structural aspects beyond the 4–7 nm pore volume. Our work aims to delve deeper into understanding catalyst-ionomer-support interactions, emphasizing the need for designing and synthesizing catalyst supports with a consistent pore size.

In addition to the catalysts support development with uniform porosity, more direct characterisation methods of the interaction between ionomers and catalysts are also required. Various methodologies have been developed to evaluate the ionomer coverage on Pt catalysts through morphological and structural characterization or electrochemical evaluation. The particle size and distribution of the catalyst aggregates could be extracted by fitting the scattering data obtained via ultrasmall angle X-ray scattering (USAXS), then to indicate the ionomer interaction change[32]. High-angle annular dark field-scanning transmission electron microscopy (HAADF-STEM) in combination with the use of Cs as a contrast enhancing dye[33], and more recently cryo-TEM have allowed direct visualisation of the location of the ionomers. Nevertheless, the accurate mapping of ionomer distribution still remains challenging, due to the low contrast between the carbon support and the ionomers as well as the possible beam damage caused by the high voltage and high electron dosage[34]. Furthermore, electrochemical evaluation provides a quantitative measurement of ionomer coverage by comparing the electric double layer capacitance or CO stripping current of the ionomer-covered area and of the total catalyst area[35]. The differentiation between ionomer-covered and non-ionomer-covered Pt surfaces was achieved by assuming that protons do not conduct to areas without ionomer coverage under dry conditions or when filled with fluorocarbon fluid. This also means that this technique can only be applied in MEA test to fulfil the required conditions[36,37]. Moreover, the coverage observed with the above methods could not give insights on the molecular interaction between the Pt and ionomer, which calls for a more close-up characterisation on the specific adsorption of ionomer ending groups on the Pt surface. Another emerging technique, namely CO displacement, was employed to quantify the adsorption of perfluorosulfonic acid ionomer by comparing the displacement charge and CO stripping charge. However, it again requires MEA setup to avoid the existence of liquid electrolyte, which would cause difficulty in interpreting the CO displacement results due to the co-adsorption of electrolyte anions and cations[38]. In contrast, X-ray absorption spectroscopy (XAS) could probe the changes caused by the adsorption of sulfonate ending group from the Nafion™ ionomer in operando with a GED setup.

On the other hand, while carbon is widely employed as the primary support for electrocatalysts due to its abundance and excellent electroconductivity, commercial carbon supports, such as carbon black and activated carbon, remain heavily reliant on petroleum sources. This reliance poses a significant challenge for the overarching goal of decarbonization in future commercial fuel cells[39]. There's a pressing need to derive carbon materials from more sustainable sources, and biomass emerges as a natural and abundant alternative. The hydrothermal carbonization of biomass presents a sustainable and economical approach to produce functional carbonaceous materials. Among the numerous sources of biomass, xylose, the most abundant pentose derived from hemicellulose, serves as a better choice than other monosaccharide such as glucose due to the more uniform morphology control offered as well as its supressed update in

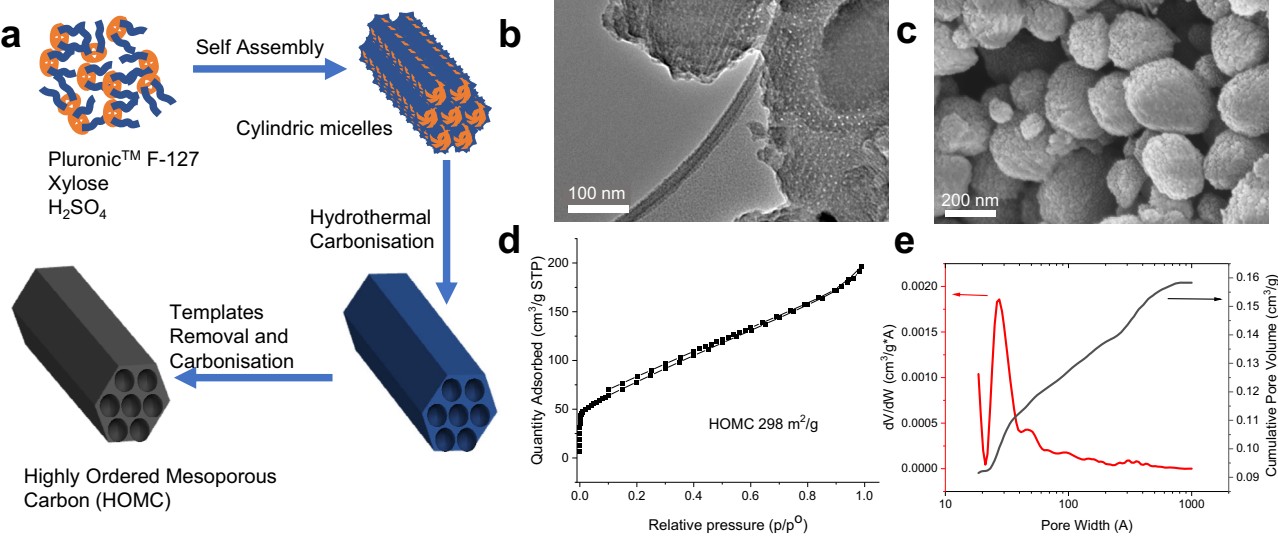

**Fig. 1 | Synthesis and characterization of highly ordered mesoporous carbon (HOMC). a** Preparation of HOMC via sulfuric acid mediated hydrothermal carbonisation of xylose. **b** TEM image of HOMC. **c** SEM image of HOMC. **d** $N_2$ sorption isotherms for HOMC at 77 K. **e** NLDFT pore size distribution for HOMC.

biological process[40]. By judiciously selecting a templating strategy using block-copolymers, mesoporous carbon with highly ordered pores can be synthesized to help understanding the structural effect of carbon support on ORR electrocatalysts.

Herein, sustainable highly ordered mesoporous carbon (HOMC) is prepared from xylose and used to rationalize the effect of carbon support morphology and porosity on Pt-ionomer interactions with a series of model Pt/C catalysts, via the use of advanced characterisation techniques. Operando XAS and gas sorption analysis are employed together with electrochemical tests in a GDE configuration, to probe the effect of mesoporosity on $O_2$ reduction activity.

## Results and discussions

### Development of highly order mesoporous carbon from biomass

Highly ordered porous carbon materials with uniform mesopores are synthesised via hydrothermal carbonisation (HTC) of xylose. As illustrated in Fig. 1a, sulfuric acid was added as the catalyst for the HTC while Pluronic™ F-127 was employed as a soft template[41]. The soft templates were removed thermally during the carbonisation of the materials to obtain highly ordered mesoporous carbon (HOMC) (Fig. 1b, c). The pore size distribution modelled with $N_2$ sorption (Fig. 1d, e) suggests uniform mesopores around 3–4 nm. In agreement with the sorption analysis, TEM images clearly show primary carbon particles about 300-500 nm size, with monodispersed 3–4 nm cylindrical pores (Fig. 1 b).

### Development of Pt/C catalysts on different carbon supports

To investigate the effect of carbon structure on the ORR performance, nanosized Pt particles were chosen as a model catalyst. In addition to the previously synthesised HOMC, two commercially widely used carbon substrates were selected, namely Ketjenblack as a highly porous carbon with high surface area and Vulcan XC-72 as a less porous carbon with low surface area. A polyol method was used to deposit Pt nanoparticles onto the different carbon supports with the Pt NPs sizes precisely controlled by the reaction pH to minimise the influence of carbon support at this stage[42]. Moreover, Pt/C catalysts were prepared with an ultra-low Pt loading to maximise the effect of carbon structure. $N_2$ sorption was carried out for all three carbon materials before and after the deposition of Pt nanoparticles. Interestingly from Fig. 2a–f, one can observe that the porosity of both Ketjenblack and Vulcan XC-72 was largely retained after the Pt loading, while 90% porosity of HOMC became inaccessible. Moreover, though all three catalysts have

similar Pt content analysed via ICP-MS (Fig. S1) after total digestion, when measuring the surface Pt content using XPS, Pt/Vulcan and Pt/ HOMC revealed similar contents of 12–14% Pt while Pt/ Ketjenblack only has about 0.5% Pt on the surface (Figs. S2 and S3, Table S1). The location of Pt on each catalyst can be inferred based on all these information. For the case of Vulcan, Pt NPs are deposited on the surface of the support, with no effect on the $N_2$ sorption-measured porosity (Fig. 2i). On Ketjenblack, the Pt NPs sit inside the pores away from the carbon surface, explaining the low Pt content measured by XPS, and the lack of perturbation to the highly porous structure (Fig. 2h). Lastly, on HOMC, Pt NPs are deposited at the surface and located at the opening of the mesoporous channels, making the mesopores no longer accessible for nitrogen sorption measurements (Fig. 2g). TEM characterisation of the catalysts suggest that the Pt nanoparticles were similar in size distribution (Figs. S4 and S5) with Pt NPs on HOMC of $2.30 \pm 0.35$ nm, Pt NPs on Vulcan of $2.19 \pm 0.51$ and Pt NPs on Ketjenblack of $2.08 \pm 0.34$ nm; however, they are located at different positions on the carbon supports. XRD of the three Pt/C catalysts were also collected (Fig. S6). Due to the low loading of Pt, the diffraction peaks specific to Pt were not detectable.

### ORR performance tested under gas diffusion electrode half-cell

The three Pt/C catalysts showed no notable difference in the ORR activity under RDE setup (Fig. S7), confirming that the Pt NPs have similar intrinsic activities. However, the ORR performance measurements done in an RDE configuration is limited by the low solubility of $O_2$ in electrolyte and thus would offer limited information on mass transport, which becomes critical at high currents. To investigate the effect of carbon support nanostructure, all ORR tests were therefore carried out in a GDE half-cell (Fig. 3a) purchased from Gasketel with in-house modification. Before testing the developed catalysts, commercial Pt/C catalysts HiSPEC4000 (40% Pt on carbon black, Johnson Matthey) was employed to benchmark the electrochemical cell. Our measurements were in good agreement with other reported works in the literature (Fig. S8). The current density in our modified cell was kept below 0.5 A/cm² for all tests to eliminate the effect of electrode and electrolyte temperature rise on uncompensated resistance ($R_u$), which are commonly observed in similar gas diffusion half cells[20]. This will also prevent the possible overestimation of the kinetics which can be induced by the variation in the temperature. According to our test (Fig. S9), no change of $R_u$ was observed by limiting the current density to 0.5 A/cm². Therefore, the cumbersome testing protocol of

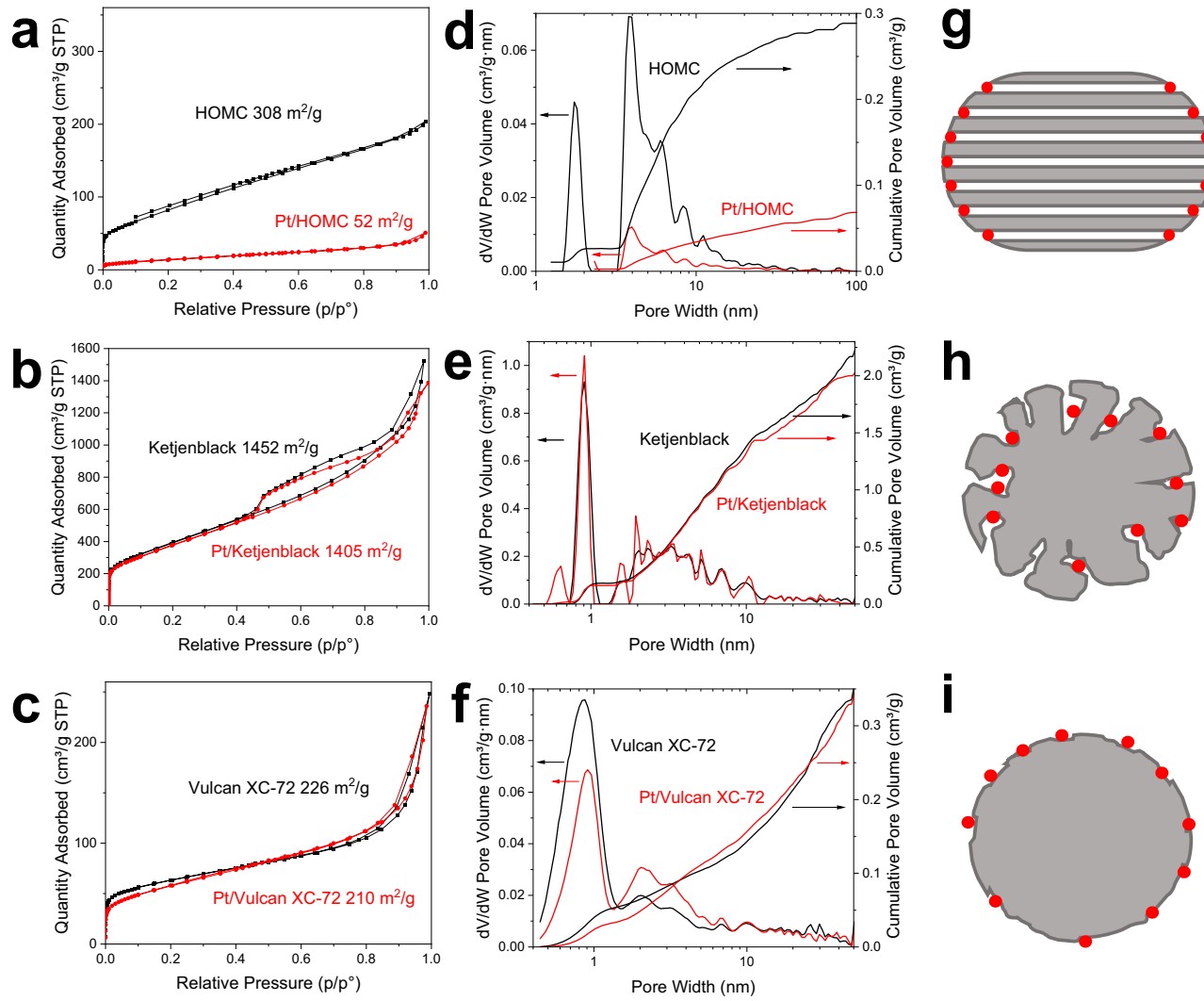

**Fig. 2 | Gas sorption analysis of carbon and Pt loaded carbon.** N₂ sorption isotherms for Pt/HOMC: (**a**), Pt/Kentjenblack (**b**) and Pt/Vulcan (**c**) at 77 K. NLDFT pore size distribution for Pt/HOMC (**d**), Pt/Kentjenblack (**e**) and Pt/Vulcan (**f**).

Models for Pt/HOMC (**g**), Pt/Kentjenblack (**h**) and Pt/Vulcan (**i**) depicting the locations of Pt nanoparticles (Pt NPs) on various catalyst supports, excluding microporosity details.

measuring resistance compensation at each potential/current point could be eliminated. Moreover, measurements were carried out in both chronoamperometry manner and linear sweep voltammetry (LSV) manner, with no difference observed between 10 mV/s LSV and 10 s holding at each potential, indicating the 10 mV/s would guarantee steady state operation of the half-cell (Fig. S10).

Gas diffusion electrodes were prepared and tested for the comparison of performance, as detailed in the supplementary information. Pt/HOMC revealed the best performance compared to Pt/Vulcan and Pt/Kentjenback (Figs. 3a, S11). As can be seen from Fig. 3a, Pt/Vulcan and Pt/HOMC showed similar behaviour at higher potential region (0.75 V – 1 V) with a slight layback of Pt/Vulcan that could be attributed to the catalyst poisoning by ionomer while more significant difference happened at high current region where mass transport effect was dominating. In contrast, Pt/Ketjenblack sustained smaller currents (when normalised to the geometric area) compared to the other two, suggesting the changed accessibility of active sites, which is also supported by its relative lower electrochemical active surface area measured with CO stripping (Fig. 3b). The ECSA values, measured via CO stripping, were 41 m²/gPt for Pt/Vulcan, 33 m²/gPt for Pt/HOMC, and 24 m²/gPt for Pt/Ketjenblack, respectively. For Pt/HOMC, the slightly lower ECSA value is attributed to the unique positioning of the

platinum nanoparticles. Predominantly located at the entrances of the mesopores, a significant match between the size of the Pt nanoparticles and the mesopore openings leads to a substantial portion of the surface atoms being obstructed by the mesoporous structure. Consequently, these atoms are rendered electrochemically inactive, impacting the ECSA. In contrast, the Vulcan XC-72 carbon support, primarily consisting of much smaller micropores (<1 nm) that cannot trap Pt nanoparticles, allows most Pt nanoparticles to be situated on the outer surface of the carbon support. This configuration results in a higher ECSA. This phenomenon is supported by the study[43], which suggests that metal nanoparticles located on the outer surface of carbon support expose more surface atoms compared to those trapped in porous structures. The ECSA of Pt/Ketjenblack is even lower due to its highly tortuous pore structure, which leads to a much higher amount of Pt nanoparticles being sealed within these pores, rendering them hardly accessible.

To further investigate the ionomer coverage across the three catalysts, the ORR current was normalized by the ECSA of each, illustrated in Fig. 3c. Notably, Pt/HOMC consistently eclipses the other two catalysts in terms of ORR current density. In the context of lower potentials, both Pt/Vulcan and Pt/Ketjenblack manifest substantial mass transport limitations in comparison to Pt/HOMC. In contrast, at a

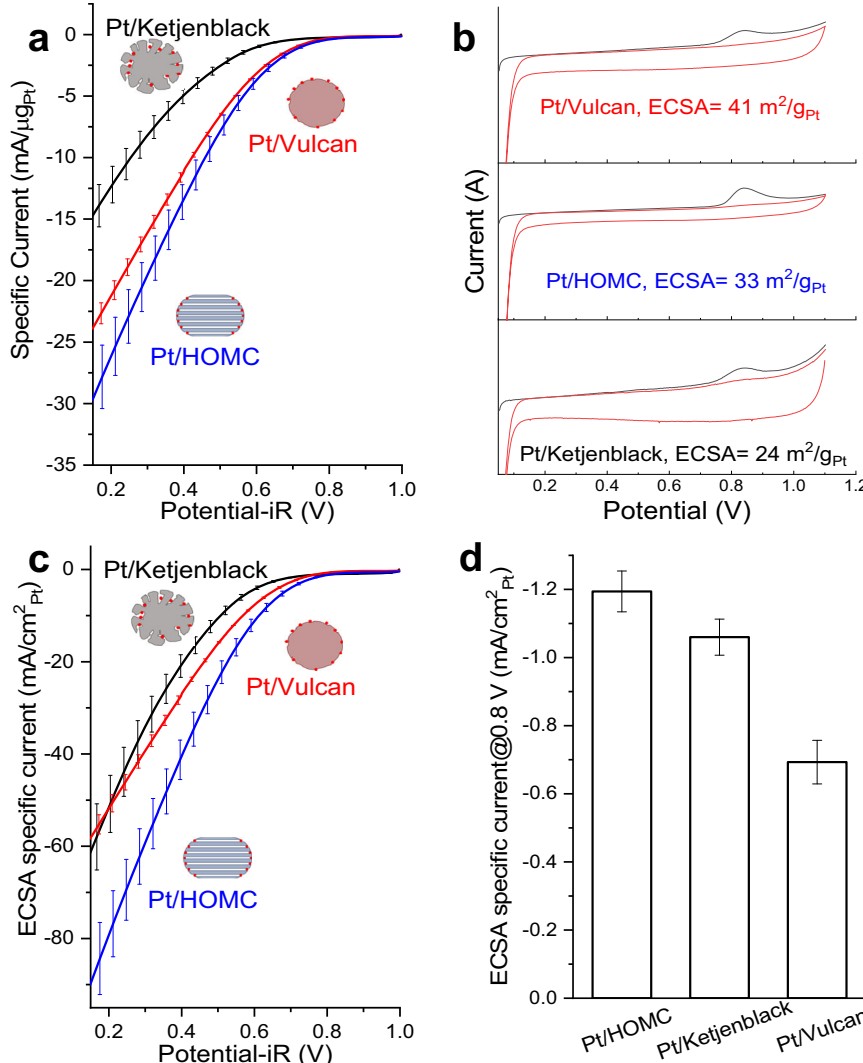

**Fig. 3 | Electrochemical performance measured under gas diffusion electrode half-cell. a** ORR polarization curves measured in gas diffusion electrode half-cell. Measurements were done in 1 M HClO₄ electrolyte at room temperature with 200 mL/min oxygen flow and 10 mV/s cathodic scanning rate. Pt wire was used as the counter electrode in the same electrolyte chamber without membrane separation, RHE was used as the reference electrode in a different electrolyte chamber connected with Luggin capillary. Potential was corrected for iR compensation with resistances of 0.87 – 1.08 Ω for all tested electrodes, as obtained from electrochemical impedance spectroscopy. Two electrodes were prepared and measured with -10 μgPt/cm² loading to generate the polarization curves. Error bar corresponds to standard deviation from two independent measurements. **b** ECSA determination via CO stripping for all three catalysts. CO stripping voltammogram (black line) and subsequent CV (red line) of the three Pt/C working electrode (-10 μgPt/cm²) recorded after the ORR measurements at a scan rate of 50 mV/s from 0.1 – 1.1 V RHE at room temperature. Potential was not iR corrected. CO was adsorbed for 5 min at 0.1 V RHE at a flow of pure CO (100 nccm) prior to the CV, followed by a 30 min purging procedure with N₂ while maintaining the potential at 0.1 V RHE. To obtain the ECSA, the CO stripping current was integrated using a linear baseline and converted to a corresponding Pt surface area using a specific capacity of 420 μC/cmPt². **c** ORR polarization curves normalised by ECSA measured with CO stripping. Error bar corresponds to standard deviation from two independent measurements. **d** ECSA specific current at 0.8 V for the three catalysts.

higher potential of 0.8 V (Fig. 3d), the current density of Pt/Vulcan markedly trails that of its counterparts, while the difference between Pt/Ketjenblack and Pt/HOMC can be neglected. This suggests that ionomer presence may impede catalyst performance via two distinct mechanisms: augmenting O₂ diffusion resistance and suppressing intrinsic activity. Subsequent sections will explore these two scenarios through gas sorption analysis and X-ray Absorption Spectroscopy.

**Ionomer-catalyst interactions**
For the mass transport effect, one important factor is the local mass transport which is largely affected by the ionomer/catalysts interaction[44,45]. The ionomer/carbon ratio is usually kept in the range of 0.5 to 1.2, while in this project, the selection of a I/C ratio of 1 has been made because it was reported to yield the highest performance in

PEMFC, when utilizing a combination of Pt/C catalysts and perfluorosulfonic acid polymer-based ionomers[46]. N₂ sorption before and after the introduction of ionomer was measured to understand the ionomer coverage (Fig. 4a–c and Figs. S12, S13). For Pt/HOMC, the introduction of ionomer predominantly affected the microporosity, as evidenced by comparisons between Fig. 4a and d, indicating that ionomer segments covered the carbon surface and blocked the micropores. Similarly, in Pt/Vulcan, the microporous structure was almost entirely obstructed due to ionomer coverage, as depicted in Fig. 4c and f. In contrast, Pt/Ketjenblack exhibited a significant alteration in porosity after the incorporation of ionomer, leading to a pronounced decrease in both micropores and mesopores (Fig. 4b and e). The blocking of micropores on the three supports were further confirmed by CO₂ sorption results (Fig. S14). The reduction in

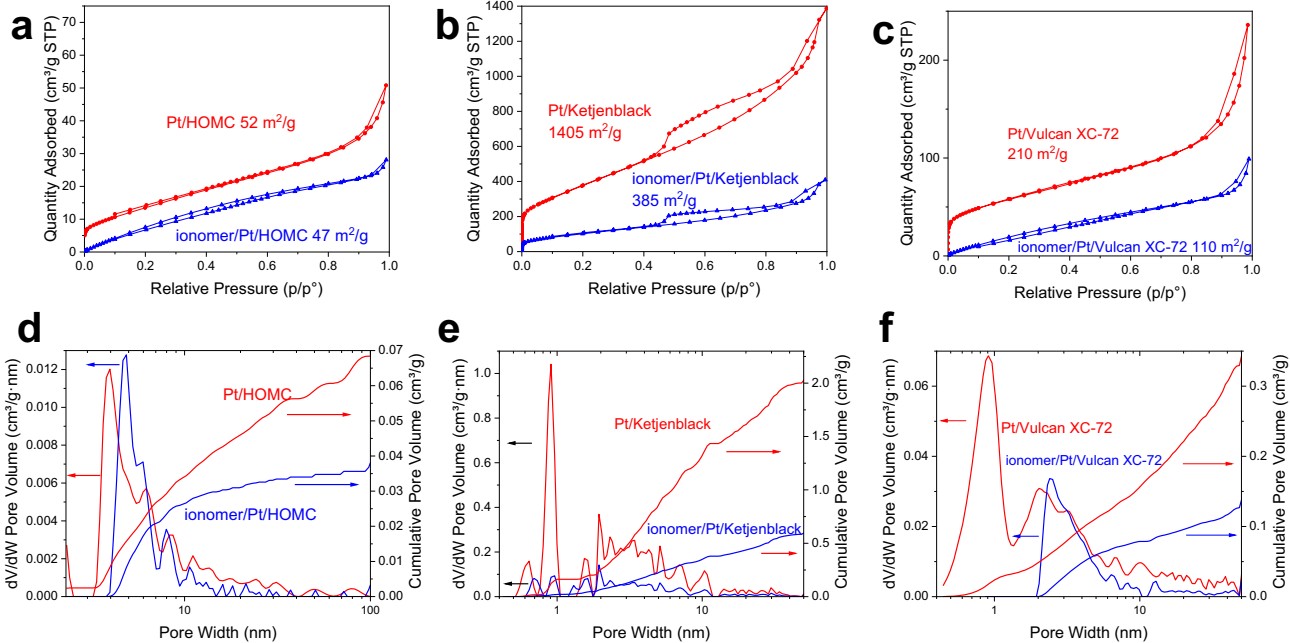

**Fig. 4 | Gas sorption analysis of catalysts with and without the ionomer.** N₂ sorption isotherms for Pt/HOMC (**a**), Pt/Ketjenblack (**b**) and Pt/Vulcan (**c**) before and after the introduction of Nafion ionomer. NLDFT pore size distribution for Pt/HOMC (**d**), Pt/Ketjenblack (**e**) and Pt/Vulcan (**f**) before and after the introduction of Nafion ionomer.

mesoporosity can be attributed to the blocking of some mesopores that have microporous openings. This interpretation is supported by the fact that the lost volume of mesopores (1.4 cm³/g_catalyst) was substantially larger than the total volume of ionomers (0.3 cm³/g_catalyst), precluding the filling of mesopores by ionomers. Subtle variations in the pore width within the mesoporous region were also observed in both Pt/HOMC and Pt/Vulcan following the introduction of ionomer. Specifically, Pt/Vulcan experienced an increase in pore size from 2.0 nm to 2.5 nm, and Pt/HOMC from 4.0 nm to 4.8 nm. These changes are attributed to modifications in adsorbate-adsorbent interactions. For Pt/HOMC, the C-BET values for the uncoated and ionomer-coated catalysts changed from 21.03 to 3.98, respectively. The significant reduction in C-BET value with ionomer coating suggests weakened interaction dynamics between nitrogen and the catalyst surface, necessitating higher partial pressures to fill the same pores. This adjustment results in a rightward shift in the adsorption isotherms, and consequently, an apparent increase in the estimated pore sizes. These observations align with findings from studies using ionic liquids and silica[47]. Despite these alterations, the mesopore openings in Pt/HOMC remain largely intact, indicating that the ionomer segments neither fill nor block the mesopore openings.

To further evidence the interaction between the ionomers and Pt NPs under ORR conditions, operando XAS analysis was carried out to investigate the specific adsorption of the ionomer under different potential over the three Pt/C. Although white line intensity change could potentially also be correlated with effect such as change in nanoparticle-support interactions, these interactions typically extend over only about one atomic distance for metallic nanoparticles[48,49]. Therefore, the majority of the Pt nanoparticles in our study would not significantly experience such interactions. On the other hand, induced by the adsorption of anions under potential, the electronic structure of Pt atoms changes accordingly. The more anions adsorbed onto the Pt surface the higher white line intensity would be expected due to the formation of Pt-O bond[50]. Kodama et al. found that the Nafion™ ionomer sulfonate groups adsorption happens at a much lower potential on Pt(111) around 0.5 V vs. RHE, comparing to OH adsorption at 0.8 V onwards vs. RHE[51]. In a related study, Ahmed et al. reported that sulfonate groups adsorption on step sites happens at potentials

<0.4 V vs. RHE, but only on those Pt electrocatalysts that possess the appropriate morphological features. These attributes include significant terrace ordering of (111), (100), or (110) facets and linear steps with a spacing of fewer than six atoms[52]. Conversely, on surfaces characterized by terraces wider than five atoms, all such electrochemical characteristics are entirely diminished, revealing only a broadened CV peak for the adsorption of sulfonate groups on Pt(111). When considering Pt NPs loaded on the three different carbon supports, it's reasonable to assume that terrace sites contain active sites for O₂ reduction and that terraces will become occupied with sulfonate at potentials positive of 0.5 V vs RHE, while OH adsorption occurs at higher potentials. This distinction in adsorption potentials facilitates the differentiation of the white line intensity increase, attributable respectively to sulfonate and OH adsorption. As shown in Fig. 5a–c and Table S2, the Pt L₃ white line intensity increased with increasing potential from 0.5 V to 1.0 V for all the three Pt/C. Same increasing trend were also observed at Pt L₂ edge (Fig. 5d–f). To distinguish the contribution of sulfonate adsorption and OH adsorption, the relative white line differences were presented in Fig. 6. A quantification method[53] (detailed in Methods section) was adopted to consider the near edge absorption differences against Pt foil reference at both L₂ and L₃ edges. This method involves the quantification of d-band vacancies in Pt catalysts by systematically analysing both L₂ and L₃ edge absorptions, with meticulous alignment, background removal and normalization across both edges, ensuring consistency and providing quantitative insights. For Pt/Vulcan XC-72, a significant increase in white line intensity at 0.6 V indicates the adsorption of sulfonate, reflecting the close contact of Pt nanoparticles with ionomers. In contrast, Pt/Ketjenblack and Pt/HOMC displayed a slight diminution in white line intensity at the same potential. This diminution is a common observation, as noted in several studies[50,54]. In these studies, such reductions in intensity are generally considered negligible and not indicative of significant changes in the platinum's electronic state. This suggests that while the diminution is observable, it should be interpreted within the inherent fluctuations typical in in situ XANES measurements. Hence, any assumption that this slight reduction indicates a substantive change in the Pt electronic state would likely be an overestimation. Following this diminution, Pt/Ketjenblack and Pt/

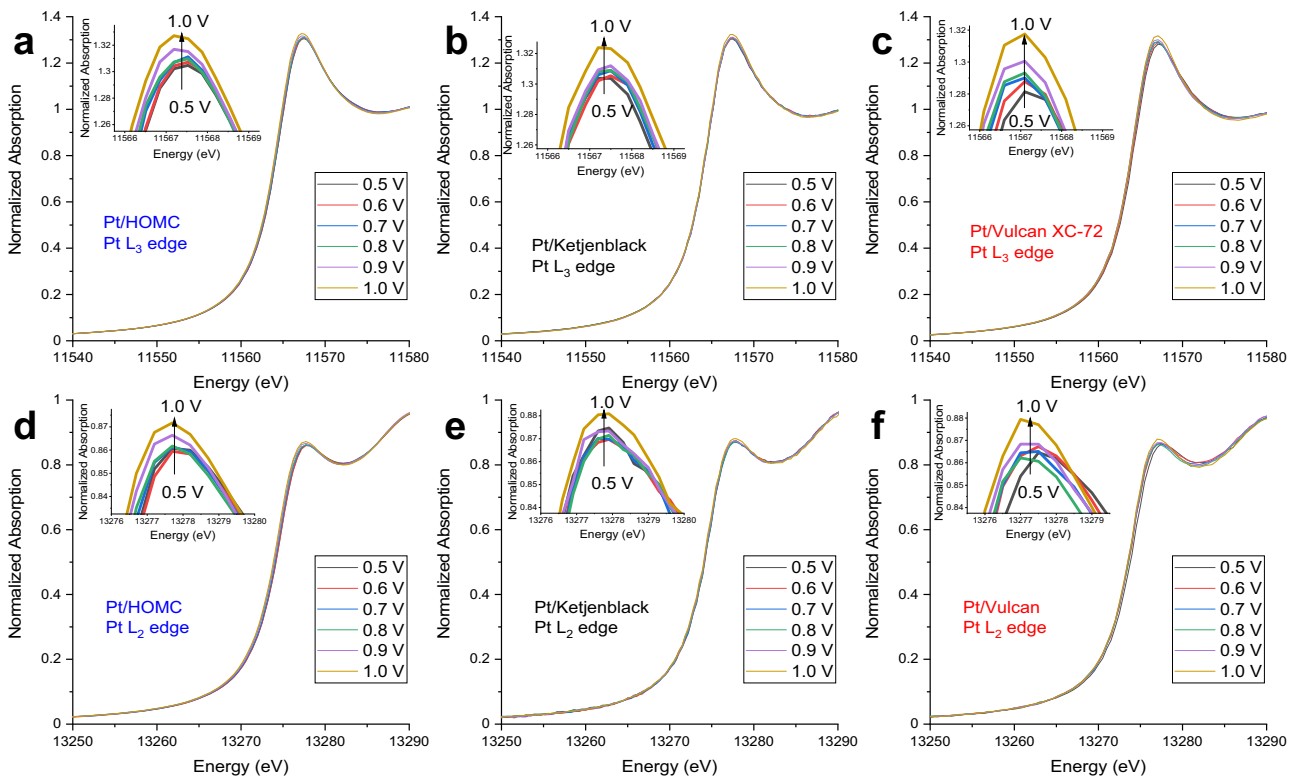

**Fig. 5 | X-ray absorption spectra.** XANES spectra at Pt $L_3$ edge for Pt/HOMC (**a**), Pt/Kentjenblack (**b**) and Pt/Vulcan (**c**); Pt $L_2$ edge for Pt/HOMC (**d**), Pt/Kentjenblack (**e**) and Pt/Vulcan (**f**).

HOMC experienced an increase in white line intensity from 0.7 V onwards, attributed to *OH adsorption. This pattern suggests that at lower potentials, no sulfonate adsorption occurs, supporting the hypothesis that most Pt nanoparticles in these catalysts, located inside the mesopores or at their mouths, are not in direct contact with the ionomers on the outer surface of the carbon support. It is also noted that the Pt/Ketjenblack shows the lowest white line intensity at 1 V, suggesting restricted -OH accessibility for this catalyst. Another important note is that the operando XAS analysis was carried out at a second synchrotron with the results suggesting the same conclusion. (Fig. S15)

Based on the $N_2$ sorption and XAS analysis, the catalyst models in Figs. 2 and 6 were developed to reveal the different ionomers-support-nanoparticles interactions and illustrated in Fig. 7. For all the three supports, ionomers deposited mostly on the outer surface blocking the micropores, but not the inside of the mesopores. Nevertheless, due to the different nature of mesopores and locations of Pt NPs, the scenarios varied in the three catalysts: (a) For Pt/Ketjenblack, the decrease in the mesoporosity could be attributed to the blocking of some mesopores with a micropore opening, because the lost volume of mesopores (1.4 $cm^3/g_{catalyst}$) was much larger than the total volume of ionomers (0.3 $cm^3/g_{catalyst}$), ruling out filling of mesopores by ionomers. The low Pt content in the outer surface, as evidenced by by XPS (Table S1), indicates a substantial portion of Pt NPs were located deep in the mesopores, having poor accessibility to $O_2$ and protons, while the blocking of some mesopores further rendered the transportation of $O_2$, leading the worst ORR performance of Pt/Ketjenblack in the three. (b) For Pt/Vulcan XC-72, there were few mesopores and the Pt NPs were on the surface under direct contact with ionomers as confirmed by the specific adsorption of sulfonate observed with XAS. However, this extra layer of ionomers added difficulty for $O_2$ diffusion as well as suppression of catalysts activity. (c) For Pt/HOMC, the opening of the mesopores were big enough and remained accessible by $N_2$ ($O_2$ under reaction condition) in the presence of ionomers.

The absence of an elevation in white line intensity at 0.6 V, which is indicative of a lack of ionomer adsorption, suggests that the Pt NPs located at the mesopore openings are not directly covered by ionomers, thereby ensuring favourable $O_2$ accessibility. This distance was nevertheless not too far, making the Pt NPs still accessible to protons from the ionomers[55]. All these optimal interactions led to the superior ORR performance of Pt/HOMC.

In conclusion, carbon nanospheres with highly ordered mesoporous channels have been synthesised sustainably from biomass. Pt nanoparticles were deposited via polyol method onto the as synthesised ordered mesoporous carbon and another two commercially widely used carbon supports (high surface Ketjenblack and low surface Vulcan XC-72). Characterization of the catalysts were carried out with SEM, TEM, BET, ICP-MS and XPS, revealing that there was no significant difference between the Pt nanoparticles in geometric size distribution and composition, indicating that the difference among these catalysts solely arose from the different support porosity, i.e. the different local environment of Pt NPs in the catalysts. All the Pt/C catalysts were tested under a gas diffusion half-cell configuration for their oxygen reduction performance, with Pt/HOMC offering the highest activity. Operando X-ray absorption spectroscopy was used to evaluate the ionomer coverage on the Pt surface by monitoring the Pt white line intensity change in the potential range of 0.5–1 V (vs RHE), and shows that among the supports tested only Vulcan suffers from sulfonate adsorption. These results, together with the comprehensive structural examination using gas sorption analysis, were used to infer the platinum and ionomer distribution on the catalyst support. Our results show that, on Vulcan, platinum and Nafion™ sit on the catalyst surface, providing good accessibility to the active sites and proton transport, but causing ionomer poisoning and extra $O_2$ transport resistance. On the contrary, platinum is deposited in the pores on Ketjenblack, while the ionomer sits on the surface. This ensures the absence of sulfonate adsorption, but causes severe proton and oxygen transport limitations, which are responsible for the low performance of this support.

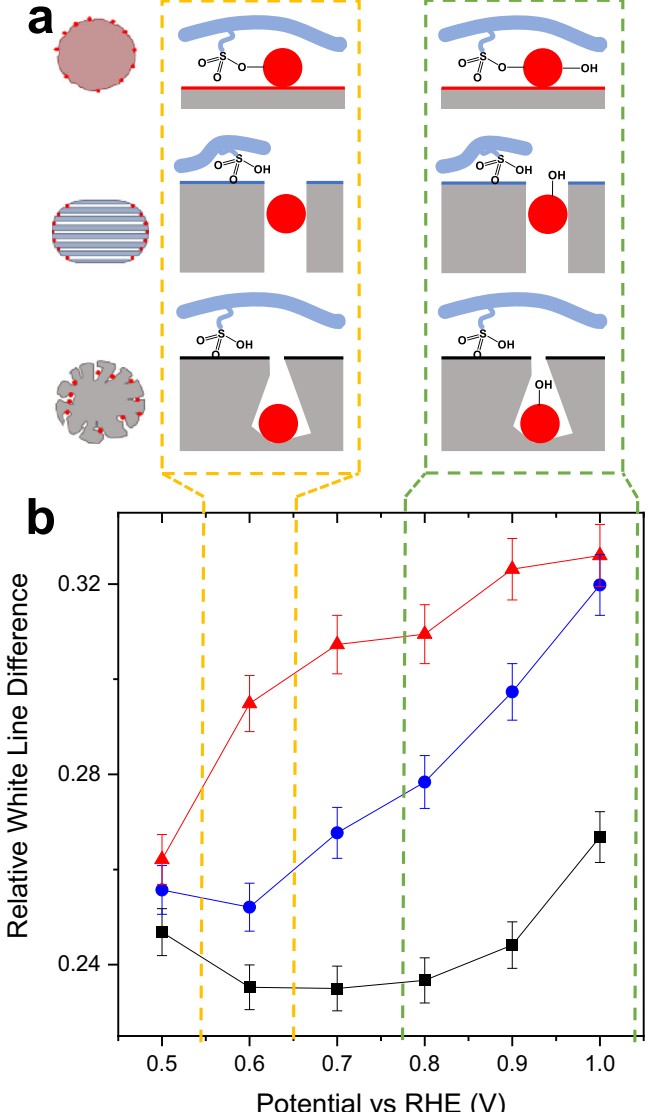

**Fig. 6 | Analysis of sulfonate and OH adsorption on Pt Catalysts. a** Schematic illustration of different ionomer interactions of the 3 catalysts. **b** Relative white line difference. Error bar corresponds to standard deviation from three independent measurements.

Finally, in the in-house synthesised HOMC, platinum sits at the entrance of the mesopores, with the ionomer closely positioned at the external surface. This explain the superior performances of HOMC, which provides more favourable interactions of Pt and Nafion™ ionomer, where direct contact in-between these two is eliminated without sacrificing the accessibility of protons for Pt active sites. This study highlights the critical role of mesoporosity in designing carbon supports for ORR catalysts and suggests avenues for further research. The superior performance of the Pt/HOMC is achieved via preferential deposition of Pt NPs at the mesopore openings, a phenomenon driven by the minimization of surface energy. This specific positioning is facilitated by the relatively short impregnation time (30 min) and the high viscosity of ethylene glycol, which limits the diffusion depth of Pt precursors into the mesoporous channels. Additionally, the polyol method allows for pH control, finely tuning the resulting Pt nanoparticle size to match the mesopore openings. These factors, combined with the intentionally designed Pt loading, ensure that Pt nanoparticles form at shallow depths at the mouths of each mesoporous channel, optimizing accessibility and reactivity. To transition

this model catalyst to practical use, development of carbon supports with a higher amount of correctly sized mesopores—to hold more Pt nanoparticles at the pore mouths—is essential. Future studies should focus on translating these findings into practical catalyst designs with higher Pt loadings. Such efforts aim to optimize the balance between catalytic performance and material costs, enhancing the viability of these catalysts in real-world applications. Future studies could benefit from CO displacement experiments to further quantify the interaction between ionomers and platinum surface, providing additional insights into catalyst behaviour under operational conditions.

## Methods

### Development of carbon materials and Pt/C catalysts

0.5 g of xylose (Sigma-Aldrich) and 1.0 g of Pluronic™ F127 (PEO$_{106}$PPO$_{70}$PEO$_{106}$, Sigma-Aldrich) were dissolved in 20 ml 10 wt% H$_2$SO$_4$ solution (prepared from 95- 98% H$_2$SO$_4$, Merck) and left stirring overnight. Then the mixture was transferred into a 30 mL Teflon-lined stainless-steel autoclave for hydrothermal carbonisation for 24 h at 150 °C. The HTC products were collected and filter-washed by water and ethanol for several times until the filtrate became clear. The carbon products were dried in an 80 °C oven overnight, after which they were carbonised at 350 °C for 1 h and then 900 °C for another 1 h with a ramping rate of 1 °C/min under N$_2$ flow of 300 mL/min to obtain HOMC.

A mixture solution was prepared by mixing 4 mL 0.2 M NaOH in ethylene glycol (anhydrous, 99.8%, Sigma-Aldrich) with 4 mL 0.004 M H$_2$PtCl$_6$ in ethylene glycol (prepared from H$_2$PtCl$_6$ · xH$_2$O (~38% Pt basis, Sigma-Aldrich)). 300 mg g of the obtained HOMC900 was then introduced and well dispersed by sonicating for 30 min. The mixture was then refluxed at 160 °C for 3 h, after which the Pt/HOMC catalysts was precipitated by washing with 0.5 M HCl solution. The same procedures were repeated using Ketjentblack EC-600 (Nouryon Chemicals) and Vulcan XC-72 (Cabot Corporation) to prepare Pt/Kentjenblack and Pt/Vulcan. All catalysts were washed with 18.2 MΩ.cm water and vacuumed dried at 80 °C for 12 h before testing.

Loading of the Pt is targeted at 1 wt% to fully utilised the mesoporous channels in HOMC based on an estimation detailed in the SI.

### Characterisations of carbon materials and Pt/C catalysts

Nitrogen isothermal adsorption/desorption was conducted on a Micromeritics 3flex system at 77 K. Before measurements, ~20 mg samples were degassed at 200 °C for 16 h under vacuum with 2 h evacuation time under the same temperature. However, for all the samples with ionomer, this degassing condition was changed to 80 °C for 16 h under vacuum with a prolonged evacuation time of 10 h, to minimize any possible effect on ionomer coverage. The ionomer containing samples were measured twice to check the reproducibility (Fig. S13). Gases used were nitrogen (N5.2 CP grade) for adsorption/desorption and He (N5.0 CP grade) for the free-space measurement. The BET equation was used to estimate the total surface area, where the fitting region was determined by the Rouquerol method on 3Flex software version 5.02[56]. The pore size distribution was calculated using slit geometry and heterogeneous surface-2D-NLDFT model in-built with 3Flex software. Regularization level was selected at the interception point of the error of fit plot and roughness of distribution plot. Carbon dioxide isothermal adsorption/desorption was conducted on a Micromeritics TriStar II Plus system at 273 K following the same degassing conditions as N$_2$ sorption. The pore size distribution was calculated using slit geometry and DFT model (CO2@273 on Carbon) in-built with TrisStar II Plus software.

XPS analysis was conducted employing a Thermo Fisher K-Alpha XPS system, and the spectra analysed with the Avantage software. All spectra were calibrated relative to the carbon C 1 s peak at 284.8 eV for correcting for charging effects.

**Fig. 7 | Ionomer-support-nanoparticle interactions in Pt/C catalysts.** Ionomer-support-nanoparticle interactions models highlighting the varying porosity of each support and its influence on the localization of Pt nanoparticles and their interactions with the ionomer.

For ICP-MS, -5 mg samples were carefully weighted to 0.1 mg accuracy and then digested in 69% $HNO_3$ (Certified AR, Eur.Ph., for analysis Fisher Chemical™, Fisher Scientific) by employing a MARS 6 microwave at 1800 W for 25 min at 215 °C. The resulting solutions were diluted to 2% $HNO_3$ for measurement against calibration standards containing Pt concentrations of 0, 25, 50, 100, 250, 400 ppb. For each sample, two duplicates were measured to get the average value.

Transmission Electron Microscopy (TEM) was carried out for the direct imaging of Pt/C catalysts in a JEOL JEM-2100F at 200 kV and the images analysed with the Gatan software. The catalysts powders were dispersed in ethanol by sonicating for 15 min and left settling for 30 min. The upper solution was collected and deposited onto the TEM grids (Holey Carbon Films on 300 Mesh Copper Grids, agar scientific). The grid was then left dry overnight and stored in dry conditions before imaging.

Operando XAS measurements were performed at room temperature using an in-house designed gas-diffusion-electrode cell placed at 45° to the incident beam. As illustrated in Fig. S17, the gas diffusion electrode was prepared similarly with air spraying onto the Freudenberg H23C8 gas diffusion layer, to obtain a total catalyst loading about 50 – 100 µgPt/cm². On one side of the GDE is a gas inlet, where oxygen or nitrogen was flowed, on the opposite side is the electrolyte (1 M $HClO_4$, prepared from ≥70% $HClO_4$ Suprapur®, Merck), containing counter electrode (platinum wire) and a capillary tube leading to the reference electrode (RHE).

Operando data were collected at B-18 (Diamond Light Source). The scanning branch of the beamline is equipped with an Si (111) scanning four bounce monochromator. The analyser crystals were aligned to the Pt L2 line and focussed onto a 4 element medipix area detector. Spectra were collected by scanning the monochromator energy over both X-ray absorption near edge structure (XANES) and extended X-ray absorption fine (EXAFS) region, with a fixed spectrometer energy. Pt foil with 25 µm thickness were used to calibrate the Si (111) double-crystal monochromator and used for data collection as a Pt standard throughout the experiments. The data was normalised to the incident intensity and processed using the Athena software package.

After CV cleaning cycles of the catalyst, the spectra were collected at the open circuit potential before the potential was stepped up from 0.5 V to 1.0 V with 0.1 V intervals. For each potential, 5 XANES were collected. The beam conditions including the beam size, incident flux at the sample, and the worst-case duration of the exposure time are summarised in Table S3. We also incorporated a method where we stepped back by 0.1 V at random intervals to verify the consistency and reliability of the data. This would also check if there were any hysteresis due to the irreversible damage or change to catalysts caused by the beam. These step-back measurements confirmed that the spectra remained consistent and reproducible, and these data were combined with the ordinary repeats for the merged spectra (see example in Fig. S18).

The edge areas were determined from the normalized data by numerical integration following Mansour's work[53]. The integration range was from −10 eV to 13 eV relating to the X-ray absorption edge, defined as the inflection point on the absorption edge step. This was done on data from both absorption edges to get $A_3$ and $A_2$ for samples, and $A_3'$ and $A_2'$ for Pt foil reference, respectively. Then the area $A_{3r}$ was

found by first calculating the difference between the areas under the Pt foil $L_3$ and $L_2$ X-ray absorption edges over the range −10 eV – 40 eV. Second, this difference in area is then multiplied by 15/14 to correct roughly for white-line weight at the $L_2$ X-ray absorption edge. The area $A_{2r}$ is estimated by multiplying this difference in area by 1/14. Finally, the relative white line difference at each potential was calculated as

$$f_d = [(A_3 - A_3') + 1.11(A_2 - A_2')]/(A_{3r} + 1.11A_{2r})$$

Pt/Vulcan and Pt/Kentjenblack were also tested at XMaS, ESRF with similar trend observed as in Fig. S15.

## Electrochemical measurements

Gas diffusion electrode (GDE) measurement was carried out in an commercial three-electrode half-cell (FlexCell-PTFE) from Gaskatel. The ink for the GDE fabrication is prepared by dispersing 4 mg catalysts and 4 mg Nafion® D-520 (5% w/w in water and 1-propanol, Alfa Aesar), in 2 ml mixture solution compromised of 50 (v/v) % isopropanol (≥99.5%, Honeywell™, Fisher Scientific) and 50(v/v) % 18.2 MΩ.cm water. The ink was homogenised firstly in an ultrasonic bath for 20 min followed by further processing with a probe sonicator (Fisherbrand model 705, USA) for another 20 min at 0 °C, under a 5 s on and 5 s off regime. GDEs were then fabricated by spraying the catalyst ink onto a Freudenberg H23C8 gas diffusion layer (230 µm thick) with an airbrush (Iwata Custom Micron CM-C PLUS) on a hot plate at 70 °C. A rubber mask was employed control the catalyst covering area to the desired area (0.5 cm², 8 mm-diameter circle). The catalysts loading of the GDEs was measure by the weight differences before and after the spray deposition.

The main and reference chamber was connected with a Haber-Luggin capillary channel and filled up with respective electrolyte for different measurements. A Pt mesh was used as the counter electrode. For acidic measurement, an in-house built RHE was used as the reference electrode. Electrochemical tests were performed without online iR correction employing an AUTOLAB PGSTAT204 coupled with 10 A booster in $N_2$ (≥99.99998% BIP® Plus, Air Products) and $O_2$ (≥99.9998% UltraPure Plus, Air Products) saturated 1 M $HClO_4$ (prepared from ≥ 70% $HClO_4$ Suprapur®, Merck) electrolyte.

Before each test, $N_2$ gas was purged to the electrolyte and the main chamber was sealed with tapes. Cyclic voltammetry was then measured for 50 cycles with $N_2$ supply in the gas chamber in the range of 0.05–1 V vs. RHE at 500 mV/s to electrochemically clean the catalysts surface. The gas was then switched to $O_2$ and the impedance spectroscopy was measured at and the open-circuit potential (OCP) in the frequency range of 10 kHz to 10 Hz to obtain the resistance for iR correction (sample spectra as in Fig. S9). Potentials were iR-compensated by manually subtracting the product of current and resistance $i \times R_u$, for which the resistance $R_u$ was obtained from a high-frequency intercept of the real resistance during electrochemical impedance spectroscopy. Lastly the linear sweep voltammetry (LSV) was performed from 1 V to 0.05 V vs. RHE at 10 mV/s. The polarisation curve was plotted after compensating the working electrode potential with equation $E = E_{measured} - iR$.

CO stripping was carried out to determine the electrochemical surface area (ECSA) of the catalysts as detailed in below diagram. The electrode was firstly cleaned under $N_2$ by cyclic voltammetry with

50 mV/s between 1.1 V and 0.1 V. After the cleaning cycles, 0.1 V vs RHE was applied to the electrode with CO (Research grade, BOC) supplied for 5 min, then the gas was switched into $N_2$ to purge the excess CO for 30 min with the same potential hold at 0.1 V. After the purging, the potential was swept from 0.1 V – 1.2 V at 50 mV/s and followed by 2 extra CV scan to testify the complete oxidation of all adsorbed CO (Fig. S19). The ECSA was estimated from the CO stripping charge by assuming the specific charge of 420 $\mu C/cm^2$.

## Data availability

Additional experimental details, experimental procedures, materials characterization, electrochemical results are available in the Supplementary Information. Source data are provided with this paper.

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

## Acknowledgements

This work is supported by Programme Grant 'Bio-derived and Bio-inspired Advanced Materials for Sustainable Industries' (grant number EP/W031019/1). We acknowledge Diamond Light Source for granting the beamtime on B18 under proposal SP29913 and thank Dr. Veronica Celorrio for her professional help. We also thank the beamtime from XMaS at ESRF under proposal 28-01-1311 and Paul B. J. Thompson and Laurence Bouchenoire for their professional help. M.W would like to thank EPSRC Centre for Doctoral Training in the Advanced Characterisation of Materials (grant number EP/L015277/1). H. L acknowledges funding from the donation by Mark Richardson to the Department of Chemical Engineering at Imperial College London. M.-M. T. thanks financial support from RAEng (CiET1819\2\60).

## Author contributions

M.W. designed and carried out the experiments. J.Z. contributed to the materials synthesis and analysed the XAS data. S.F., H.L. and L.J.R.H. contributed to the XAS measurements. I.E.L.S and M.-M.T. supervised the project. M.W. wrote the manuscript with input from J.Z., S.F., H.L., L.J.R.H., I.E.L.S and M.-M.T.

## Competing interests

The authors declare no competing interests.
