## [Peer Review File · Nature Communications]

REVIEWER COMMENTS

Reviewer #1 (Remarks to the Author):

In this submitted paper, Pt/Vulcan XC-72, Pt/Ketjenblack and Pt/HOMC were prepared and evaluated by Operando X-ray absorption spectroscopy, BET, TEM, XPS and ICP. Based on the abovementioned characterization, the local environment of Pt NPs in the porous carbon support and corresponding effect stemmed from ionomer coverage are studied. Undoubtedly, the reported results are meaningful for the development of Nafion ionomer modified electrode. However, there are still some unclear mechanism and confusing expression in the manuscript. In my opinion, this work should be major revised after addressing the following comments.

Here are some comments for the authors:

1. Both the Figure 1d and 2a display the N₂ sorption isotherm of HOMC. But one is 270 m²/g and the other is 308 m²/g. Why the data is inconsistent? It because the synthesis of HOMC has poor reproducibility, or errors in using data. Author should carefully check the data and Figures in the manuscript, based on the respect for journal.
2. XPS survey of Pt/Ketjenblack, Pt/Vulcan and Pt/HOMC is suggested to be given. And, the Pt contents tested by XPS are usually shown in form of molar ratio. So, authors should carefully check it.
3. The structure model in Figure 2i is illogical and ridiculous. The BET results of Vulcan XC-72 and HOMC is similar. Why the structure of Vulcan is nearly nonporous? Authors should ensure the logicity of the structure model.
4. Authors said a small amount of Pt atoms facing towards the inaccessible mesoporous channels could be inaccessible, thus decreasing the weight-normalized ECSA of Pt. It is quite confusing. The Pt atoms facing towards inaccessible nonporous carbon of Vulcan XC-72 is also inaccessible. Why Pt/Vulcan has a higher ECSA value than Pt/HOMC ?
5. Authors used N₂ sorption isotherms to evaluate the mass transport effect influenced by ionomer. To my knowledge, the sample should be pretreated by heating to high temperature before N₂ sorption tests, in order to remove the absorbed impurities. Authors also said the pretreatment process in the section that characterisations of carbon materials and Pt/C catalyts. Thus, how can authors ensure that the pretreatment process has little effect on the ionomer coverage? The results of repeated tests on Pt/HOMC, Pt/Ketjenblack and Pt/Vulcan after the introduction of Nafion ionomer should be given.
6. Why the ionomer film covers the mesoporous of Pt/Ketjenblack, but keep away from the mesoporous of Pt/HOMC?
7. It is confusing that, the variation in mesoporous region (above 3 nm) of Pt/HOMC (Figure 4d) demonstrate a distinct effect triggered by the introduction of Nafion ionomer. Why authors draw an unaffected structure in Figure 6a. And, if no sulfonate adsorption at low potential range on Pt/Ketjenblack and Pt/HOMC, why the diminution in the white line intensity is inconsistent? And what is the reason for the diminution in the white line intensity at 0.6 V?
8. For the section of conclusion, author briefly summarize the results of deposited platinum on diverse carbon support for the influence of ionomer coverage. However, it is just the assumption and explanation based on the results. Authors should give a more helpful viewpoint for the reason why the Pt nanoparticles can locate at the mesopore mouths, which is significant for the fabrication of highly active electrocatalyst.

Reviewer #2 (Remarks to the Author):

In this manuscript, Wang et al demonstrate how carbon support microstructure can control the properties of a fuel cell catalyst, and furthermore control ionomer structure. This is an important topic relevant to many types of electrocatalysts, and it is a famously difficult field of experimentation. Catalyst model systems using three different types of carefully selected carbon support were fabricated, which demonstrate significant changes in gas sorption properties. A new variant of operando X-ray spectroscopy cell is described and benchmarked, and an operando XANES measurement is shown, although the connection between the X-ray and gas sorption work are not very strong. This manuscript has a few substantial issues, which if corrected would make this a nice study of short length suitable for publication in a high impact journal.

In the introduction, it is fair to be a bit harsher on the various electrochemical methods for probing ionomer interactions. While calling them 'quantitative' is technically accurate, I just wish we knew what we are quantifying! There is virtually no structural evidence for the necessary assumptions. For example, the "dry" CO stripping measurement likely responds to more than just the Pt in contact with ionomer, for a variable definition of "in contact". That said, I was quite surprised not to see the "CO displacement" experiment run in this work. Unlike the CO stripping, you polarize the cell and then flow CO in, generating a transient that allows the charge of desorbing species at the interface to be integrated. This technique is now rather popular, and is capable, in theory, of calculating the ionomer coverage on Pt catalyst surfaces under relevant conditions.

The choice of extremely low 1wt% Pt loading is interesting in the sense of building a model system, but limits the extension of this model to practical catalysts where 30-70 wt% Pt are routinely employed. I admit I did not really understand what the authors were trying to do here the first time I looked through the paper, and was ready to criticize this unrealistic feature. Part of my comprehension issue arised from the concept/objective of this work not being introduced earlier in the introduction, and only shows up mid way through the results and discussion section, where it is not connected to the literature review. One question of these low loadings is whether the polyol method selectively deposits Pt at specific points on the carbon support. These high energy adsorption sites may strongly influence the pore structure and nature of the blocking, which is unique to the author's system.

The gas sorption measurements are a little puzzling to me. Better analysis of these curves could be quite informative towards understanding the microstructure of the HOMC, which is at the heart of this manuscript. No doubt the HOMC carbon has decent surface area and some porosity, visible in the TEM. But for a paper that relies very strongly on a conceptual model like the cartoons in Fig 2, the question becomes how correct these are.

Firstly, I don't see the traditional capillary condensation and hysteresis for HOMC I would normally expect for open mesopores. My understanding was that closed or severely bottlenecked pores produce

less, and more complex hysteresis, while open cylindrical pores like those drawn in Figs 1 and 2 should give a very pronounced hysteresis with a classically shaped isotherm. When I look at the sorption data in Fig 1 of the manuscript, the curves are not labelled, but I assume the one starting at low P/P_0 is the sorption curve. The desorption curve actually crosses back underneath the sorption curve, which I would like a physical explanation for. The common errors I have seen, like setting too short of an equilibration time, usually result in undershooting the sorption curve, and increasing hysteresis, not the inverse. Usually I see DFT methods used to calculate the pore size distribution from the desorption half of the hysteresis loop, although adsorption methods can also be used. I did not understand which parts of which curves were included in this model. The P/P_0 log plot was not included in the SI, so I could not review the low pressure data. Confusingly the adsorption data of the bare HOMC in Figure 1 does not match the adsorption/desorption data in Figure 2a, even though as far as I can tell, this is the same material. In contrast to Figure 1, the data in Figure 2 appears to be exceptionally high quality. For example, Fig 2b shows the H3 style isotherm characteristic of wedge-shaped pores in Ketjen Black which is absolutely textbook, and in line with the cartoon.

Creative control experiments may significantly enhance the credibility of the cartoon model. N₂ isotherms performed at 77K are extremely sensitive to pore blocking. Other gases at higher temperature are usually employed to evaluate whether pores are actually capped, and the difference even with an argon isotherm at 87K can be dramatic. Since PEMFCs do not operate at 77K, there is some thought needed to the validity of this aspect of the model. More precise language may also help a reader here. For example, after Pt deposition it is claimed that nearly 90% of the HOMC porosity is lost. However, this porosity still exists, and is merely inaccessible under the measurement conditions, which is mentioned later on in the discussion. Grinding or lightly milling the Pt/HOMC could break the support particles into pieces, reopening the sealed pores and restoring the enormous original surface area.

That Yargaladda paper discussed in the introduction is frustrating, since it does not show any data like TEM or isotherms or electrochemistry. It also does not say how the catalyst films are actually made or deposited, the parameters of which play a massive role in the performance of the device. That paper, and its very high number of citations is a good example of the type of sparsely characterized work being done in the field, which the present authors are able to surpass using high quality data.

The CVs in Figure 3 do not show any H-UPD phenomena, or even clear Pt oxidation, which implies they are very strongly contaminated even after one sweep to high potential. High quality, clean CVs are the essential component for evaluating ORR activity and ECSA of Pt based PEMFC electrocatalysts. While the high Pt to carbon ratio will make the capacitance a bit larger than usual, we can estimate the size of these Pt specific features from the CO stripping peaks, and see they are absent. This is quite a show-stopper, and I cannot recommend publication without it, unless comparisons of catalytic activity and ECSA are removed. I recommend the authors activate their catalysts by cycling 0.05-1.2V at perhaps 100mvps for maybe 25 cycles or until stable, to clean the surface before attempting any sort of CO sorption or ORR. To obtain the ECSA with CO stripping, it is best to subtract the subsequent, CO-free CV from the CO-bound one, instead of trying to use a linear baseline correction. Of course there is always discussion of what activation does to catalyst structure, but without understanding what state the Pt is in, going the next step to probing the Pt-ionomer interactions seems impossible.

Unfortunately the same limitations about not knowing what the surface chemistry is like also applies to the XANES experiment, although the standards for operando cells are much lower. Increasingly high potentials will slowly clean the surface over time. In addition, we can't discount the influence of the X-ray beam, which generates considerable quantities of radicals and peroxides which clean loosely bound organic residues, and ionomer sidechain degradation products, which are powerful poisons known to bind very tightly to the Pt surface. Was any check made on the reversibility of the XANES spectra, like stepping back to 0.5V after the spectra at 1.0V was collected? Was a spectra collected under CO atmosphere, or was this too unsafe? White line intensity is an extremely flexible parameter, in that various authors routinely invoke small shifts to be correlated with any convenient structural change du jour, such as the strength of particle-support interaction, which also changes with potential. I was surprised not to see the particle support interaction mentioned at all, given the comprehensive XAS work on all the catalysts. I do quite like the XANES idea, but additional discussion re: the expected change in intensity would be helpful in supporting the interpretation, since so many different things could be happening. Is there a specific reason that entire spectra were collected? Why not just measure 1 point pre and post edge for normalization, and one at the max intensity? This would presumably allow for much faster data acquisition, and even operando XANES during cyclic voltammetry would be possible.

The benchmarking of the half cell was obviously performed with a high degree of care, and the detailed descriptions and controls, eg checking for steady state conditions, is greatly appreciated.

It would seem that electron tomography (10.1017/S1431927617011114) or FESEM (10.1021/acsaem.2c01790) are much better than regular TEM to understand whether catalysts are buried inside the carbon, where the 2D projected image provides very limited depth information.

There are a few spots where the typo Kentjen is used instead of Ketjen.

Reviewer #3 (Remarks to the Author):

The investigation into the effects of carbon support pore size on particle dispersion, catalyst-ionomer interaction, and cell performance has garnered significant attention. As outlined in the introductory section, the elusive nature of these theories is often attributed to the tortuous pore structure of current carbon supports and the limitations of characterization techniques. Wang et al. addressed this challenge by employing three distinct carbon supports with varying pore distributions to examine the correlation between pore size and catalyst-ionomer interaction. This study utilized operando X-ray absorption spectroscopy and gas sorption analysis techniques. Despite the advanced nature of the XAS method, the results cannot distinctly differentiate among the three types of support because of the inconsistent intrinsic activities of Pt particle and yielding no novel conclusions. The following issues need addressing before further consideration:

1. The manuscript contains typos, such as the TEM comparison in Figure S3 instead of Figure S1 (line 172).
2. Pt loading in the samples is extremely low (<1 wt%) based on ICP data, yet the Pt particle size is not small and is similar to commercial Pt/C. Clarification is needed on how this low loading with normal-sized Pt particles reduces HOMC support porosity by 90%.
3. The quality of TEM in Figure S3 is subpar. The figures should have the same magnification bar, as the current representation suggests a higher Pt loading for Pt/HOMC, with Pt particles mostly aggregating around the surface of the HOMC support.
4. It is better to provide XRD data to estimate the particle size of the three catalyst types.
5. ORR activity comparison should be provided, evaluated under the liquid cell using the Glassy carbon rotation disk electrode. Excluding ionomer or transport impacts, the intrinsic Pt activities of the three catalysts should be demonstrated as equal.
6. An explanation is required for why Pt/Ketjenblack exhibits a much lower ECSA and poorer performance than Pt/Vulcan or Pt/HOMC in Figure 3, even at high potential regions. These data seem inconsistent with literature results, indicating potential issues with the Pt dispersion approach.
7. With the addition of the same amount of ionomer, why only the surface area of Pt/Ketjenblack experiences around a 73% decrease, as the ionomer film also constructed on Vulcan or HOMC supports?
8. The rationale for the increase in pore width of ionomer-coated Pt/Vulcan and Pt/HOMC compared to the uncoated situation needs to be elucidated.
9. Ex-situ XAS comparison for the catalysts with and without ionomer addition should be provided to analyze potential discrepancies in ionomer distribution.
10. A table illustrating the white line intensity change concerning potential variations for the three catalysts should be included for clearer comparison.

Response to Reviewers for Nature Communications (NCOMMS-23-58195)

The comments by the reviewers are in grey italic text and the authors' replies are in green text, with revisions to the manuscript highlighted in yellow.

Reviewer #1 (Remarks to the Author):

In this submitted paper, Pt/Vulcan XC-72, Pt/Ketjenblack and Pt/HOMC were prepared and evaluated by Operando X-ray absorption spectroscopy, BET, TEM, XPS and ICP. Based on the abovementioned characterization, the local environment of Pt NPs in the porous carbon support and corresponding effect stemmed from ionomer coverage are studied. Undoubtedly, the reported results are meaningful for the development of Nafion ionomer modified electrode. However, there are still some unclear mechanism and confusing expression in the manuscript. In my opinion, this work should be major revised after addressing the following comments.

Response:

Thank you for your insightful comments and for recognizing the meaningful contribution of our work. We appreciate your suggestions for clarifying certain mechanisms and expressions within the manuscript. We have addressed these points and we do hope that the clarity and impact of our study has improved. Below, we have addressed each of your specific comments and made corresponding revisions to the manuscript.

1. Both the Figure 1d and 2a display the N₂ sorption isotherm of HOMC. But one is 270 m²/g and the other is 308 m²/g. Why the data is inconsistent? It because the synthesis of HOMC has poor reproducibility, or errors in using data. Author should carefully check the data and Figures in the manuscript, based on the respect for journal.

Response:

We apologize for the oversight in not addressing the inconsistencies between the data presented in Figures 1 and 2. During the initial phase of this project, gas sorption measurements were conducted using the Tristar Plus II instrument from Micromeritics, which provided the data shown in Figure 1. As the project progressed, we transitioned to the more advanced 3Flex instrument from Micromeritics, known for its superior control and resolution of lower pressure measurements. Consequently, the data presented in Figures 2 and 4 exhibit higher quality and resolution. To ensure consistency and accuracy, we have repeated the gas sorption measurements for bare HOMC using the 3Flex instrument. The new measurements are consistent with the high-quality data shown in Figure 2 and the surface area obtained was 298m²/g compared to 308 m²/g in the previous test, which we believe is acceptable. Accordingly, we have updated Figure 1 with the new data obtained using the 3Flex instrument. We appreciate the reviewer's attention to this detail, which has helped us improve the clarity and reliability of our manuscript.

Action taken:

Updated Figure 1 in Manuscript:

2. XPS survey of Pt/Ketjenblack, Pt/Vulcan and Pt/HOMC is suggested to be given. And, the Pt contents tested by XPS are usually shown in form of molar ratio. So, authors should carefully check it.

Response:

Thank you for your constructive suggestion. We have included these spectra in the Supplementary Information, providing comprehensive insights into the surface composition of each catalyst.

Regarding the presentation of Pt content, we acknowledge your point on displaying the data in the form of molar ratios. In our study, the platinum content initially determined by XPS is indeed given as an atom ratio. We have clarified in the caption of Table S1 that the reported weight ratios were derived from these atom ratios using standard atomic weights, ensuring the accuracy and relevance of our data presentation.

Action taken:

Added following in SI:

Figure S2. XPS survey of the Pt/C catalysts.

Table S1. Bulk Pt content measured via ICP and surface Pt content measured via XPS for the 3 catalysts. The weight ratios were derived from XPS atom ratios using standard atomic weights.

	Pt/Ketjenblack	Pt/Vulcan	Pt/HOMC
Pt content by ICP	0.58 wt%	0.78 wt%	0.75 wt%
Surface Pt content by XPS	0.49 wt%	14.98 wt%	12.6 wt%

3. The structure model in Figure 2i is illogical and ridiculous. The BET results of Vulcan XC-72 and HOMC is similar. Why the structure of Vulcan is nearly nonporous? Authors should ensure the logicity of the structure model.

Response:

Thank you for your comments regarding the illustration of the structural model in Figure 2i. We acknowledge the differences in the BET isotherms of Vulcan XC-72 compared to Ketjenblack and HOMC, as highlighted in the referenced literature (DOI: 10.1039/c9cc08066j for HOMC, and <https://doi.org/10.1016/j.electacta.2014.04.080> for Vulcan XC-72). In light of these references, we confirm that Vulcan XC-72 is characterized by its exclusive microporosity, which contrasts with the presence of mesoporosity in Ketjenblack and HOMC.

The purpose of the model in Figure 2i is specifically to illustrate the absence of mesoporosity in Vulcan XC-72, a key differentiation from the other materials. This distinction is crucial to the arguments we present later in the manuscript. The discussion and visualization of microporosity across the materials, as correctly noted, are extensively covered in Figure 7.

To ensure that this distinction is clear and to avoid any potential misunderstanding, we have revised the manuscript to provide a clearer explanation in the figure captions of Figures 2i and 7, highlighting the specific focus of each figure within the context of our study's broader narrative.

Action taken:

Revision in Manuscript:

Figure 2. N₂ sorption isotherms for Pt/HOMC (a), Pt/Ketjenblack (b) and Pt/Vulcan (c) at 77K. NLDFT pore size distribution for Pt/HOMC (d), Pt/Ketjenblack (e) and Pt/Vulcan (f). **Models for Pt/HOMC**

(g), Pt/Kentjenblack (h) and Pt/Vulcan (i) depicting the locations of Pt nanoparticles (Pt NPs) on various catalyst supports, excluding microporosity details.

Figure 7. Ionomers-support-nanoparticles interaction models for the three Pt/C catalysts, highlighting the varying porosity of each support and its influence on the localization of Pt nanoparticles and their interactions with the ionomer.

4. Authors said a small amount of Pt atoms facing towards the inaccessible mesoporous channels could be inaccessible, thus decreasing the weight-normalized ECSA of Pt. It is quite confusing. The Pt atoms facing towards inaccessible nonporous carbon of Vulcan XC-72 is also inaccessible. Why Pt/Vulcan has a higher ECSA value than Pt/HOMC?

Response:

Thank you for your insightful comment. We appreciate the opportunity to clarify the specifics of the ECSA values for Pt/Vulcan and Pt/HOMC.

Our findings from TEM, N₂ sorption and XPS indicate that for Pt/HOMC, the platinum nanoparticles are predominantly located at the entrances of the mesopores. This unique positioning, coupled with the size match between the Pt nanoparticles and the mesopore openings, leads to a significant portion (~50%) of the surface atoms being obstructed by the mesoporous structure. As a result, these atoms are rendered electrochemically inaccessible, which significantly impacts the ECSA.

In contrast, the Vulcan XC-72 carbon support primarily consists of much smaller micropores (<1 nm) that cannot trap Pt nanoparticles. Consequently, most Pt nanoparticles are located on the outer surface of the carbon support, leading to a higher ECSA. This is supported by the study (DOI: 10.1039/D3SC04691E), which suggests that metal nanoparticles located on the outer surface of carbon support expose more surface atoms compared to those trapped in porous structures.

To ensure the manuscript accurately conveys this phenomenon, we have included a revised discussion in the text to clarify the positioning and accessibility of Pt nanoparticles on different carbon supports.

Action taken:

Revision in Manuscript:

The ECSA values, measured via CO stripping, were 41 m²/gPt for Pt/Vulcan, 33 m²/gPt for Pt/HOMC, and 24 m²/gPt for Pt/Ketjenblack, respectively. For Pt/HOMC, the slightly lower ECSA value is attributed to the unique positioning of the platinum nanoparticles. Predominantly located at the entrances of the mesopores, a significant match between the size of the Pt nanoparticles and the mesopore openings leads to a substantial portion of the surface atoms being obstructed by the mesoporous structure. Consequently, these atoms are rendered electrochemically inactive, impacting the ECSA. In contrast, the Vulcan XC-72 carbon support, primarily consisting of much smaller micropores (<1 nm) that cannot trap Pt nanoparticles, allows most Pt nanoparticles to be situated on the outer surface of the carbon support. This configuration results in a higher ECSA. This phenomenon is supported by the study³⁹, which suggests that metal nanoparticles located on the outer surface of carbon support expose more surface atoms compared to those trapped in porous structures. The ECSA of Pt/Ketjenblack is even lower due to its highly tortuous pore structure, which

leads to a much higher amount of Pt nanoparticles being sealed within these pores, rendering them hardly accessible.

5. Authors used N₂ sorption isotherms to evaluate the mass transport effect influenced by ionomer. To my knowledge, the sample should be pretreated by heating to high temperature before N₂ sorption tests, in order to remove the absorbed impurities. Authors also said the pretreatment process in the section that characterisations of carbon materials and Pt/C catalysts. Thus, how can authors ensure that the pretreatment process has little effect on the ionomer coverage? The results of repeated tests on Pt/HOMC, Pt/Kentjenblack and Pt/Vulcan after the introduction of Nafion ionomer should be given.

Response:

Thank you for your detailed inquiry regarding the pretreatment process applied before conducting N₂ sorption isotherms and its potential impact on ionomer coverage. We appreciate the opportunity to clarify this aspect of our methodology.

In our study, the pretreatment of samples of bare carbons and Pt loaded carbons were done at 200 °C for 16 h under vacuum with 2 h evacuation under the same temperature. However, for all the samples with ionomer, this degassing condition was lowered to 80°C for 16 h under vacuum with a prolonged evacuation time of 10h. This temperature is close to the typical operating temperatures of PEMFCs and well below the melting temperature of Nafion ionomer. Such conditions ensure that while impurities are effectively removed, the structural integrity and distribution of the Nafion ionomer remain unaffected. Also, the overlapping of the adsorption and desorption curves demonstrates that the lowered degassing condition was sufficient.

To further validate that the degassing process had minimal impact on the ionomer coverage, we provided the repeated tests results. The results showed consistent BET surface area and porosity, confirming that our pretreatment process preserves the functional characteristics of the ionomer within the catalyst structure.

These findings, including detailed experimental data and comparative analysis, have been incorporated into the revised manuscript in the section on characterizations of carbon materials and Pt/C catalysts. This additional information provides a robust basis for the reliability of our N₂ sorption measurements following the described pretreatment.

Action taken:

Added text in Manuscript:

Before measurements, ~20 mg samples were degassed at 200 °C for 16 h under vacuum with 2 h evacuation time under the same temperature. However, for all the samples with ionomer, this degassing condition was changed to 80°C for 16 h under vacuum with a prolonged evacuation time of 10h, to minimize any possible effect on ionomer coverage. The ionomer containing samples were measured twice to check the reproducibility (Figure S13).

Added figure in SI:

Figure S13. N₂ sorption of repeated tests for ionomer-containing samples.

6. Why the ionomer film covers the mesoporous of Pt/Ketjenblack, but keep away from the mesoporous of Pt/HOMC?

Response:

Thank you for your question regarding the differential ionomer coverage between Pt/Ketjenblack and Pt/HOMC. We have addressed this observation in detail within our manuscript, specifically in the discussion related to N₂ sorption, XAS analysis, and the catalyst models presented in Figures 2, 6, and 7.

As outlined in our manuscript, the ionomer predominantly deposits on the outer surfaces of all three supports, effectively blocking micropores without filling the mesopores. However, the nature and location of mesopores and Pt nanoparticles lead to varying scenarios across the different catalysts:

a) Pt/Ketjenblack experiences a significant decrease in mesoporosity due to the ionomer blocking mesopores with micropore openings. The considerable discrepancy between the lost mesopore volume ($1.4 \text{ cm}^3/\text{g}_{\text{catalyst}}$) and the total ionomer volume ($0.3 \text{ cm}^3/\text{g}_{\text{catalyst}}$) supports the conclusion that mesopores are not filled but are blocked. XPS analysis reveals a low Pt content on the outer surface, suggesting that many Pt nanoparticles are located deeper within the mesopores, which limits O₂ and proton accessibility, subsequently affecting ORR performance.

b) Pt/Vulcan XC-72 has fewer mesopores, and the Pt nanoparticles are situated on the surface in direct contact with ionomers. The presence of an additional ionomer layer, confirmed by the specific adsorption of sulfonate via XAS, complicates O₂ diffusion and suppresses catalyst activity.

c) Pt/HOMC features larger mesopore openings that remain accessible to N₂ (and O₂ under reaction conditions) despite the presence of ionomers, indicating minimal interference with gas transport.

These detailed explanations are aimed at clarifying the differential effects of ionomer coverage on the mesoporosity and accessibility of the catalysts.

Action taken:

Revision in Manuscript:

N₂ sorption before and after the introduction of ionomer was measured to understand the ionomer coverage (Figure 4a-c & Figure S12, S13). For Pt/HOMC, the introduction of ionomer predominantly affected the microporosity, as evidenced by comparisons between Figures 4a and 4d, indicating that ionomer segments covered the carbon surface and blocked the micropores. Similarly, in Pt/Vulcan, the microporous structure was almost entirely obstructed due to ionomer coverage, as depicted in Figures 4c and 4f. In contrast, Pt/Ketjenblack exhibited a significant alteration in porosity after the incorporation of ionomer, leading to a pronounced decrease in both micropores and mesopores (Figures 4b and 4e). The blocking of micropores on the three supports were further confirmed by CO₂ sorption results (Figure S14). The reduction in mesoporosity can be attributed to the blocking of some mesopores that have microporous openings. This interpretation is supported by the fact that the lost volume of mesopores ($1.4 \text{ cm}^3/\text{g}_{\text{catalyst}}$) was substantially larger than the total volume of ionomers ($0.3 \text{ cm}^3/\text{g}_{\text{catalyst}}$), precluding the filling of mesopores by ionomers.

7. It is confusing that, the variation in mesoporous region (above 3 nm) of Pt/HOMC (Figure 4d) demonstrate a distinct effect triggered by the introduction of Nafion ionomer. Why authors draw an unaffected structure in Figure 6a.

Response:

We appreciate the reviewer's detailed observations on the pore width variations in ionomer-coated Pt/HOMC catalysts compared to their uncoated counterparts, a phenomenon similarly observed in Pt/Vulcan. Notably, the shift in the peak position is relatively small, with Pt/Vulcan showing an increase from 2.0 nm to 2.5 nm, and Pt/HOMC from 4.0 nm to 4.8 nm. Such changes suggest subtle modifications in the pore structure upon ionomer coating.

The observed alterations in pore width are attributed to changes in adsorbate-adsorbent interactions. Specifically, for Pt/HOMC, the C-BET values for the uncoated and ionomer-coated catalysts are 21.03 and 3.98, respectively. The decrease in the C-BET value with ionomer coating indicates weakened interaction dynamics between nitrogen and the catalyst surface. This weaker interaction could necessitate higher partial pressures to fill the same pores, possibly leading to a rightward shift in the adsorption isotherms and consequently, an apparent increase in the estimated pore sizes.

The relevance of these observations is supported by analogous phenomena reported in systems utilizing ionic liquids and silica, as detailed in the referenced study (doi.org/10.1039/C4CP02749C). This supports our interpretation that ionomer coating modifies surface interactions, thereby influencing pore size characterization.

Regarding the unaffected mesoporous structure depicted in Figure 6a for Pt/HOMC:

The illustration of an unchanged mesoporous structure was intended to highlight the relative stability of the mesopore openings in Pt/HOMC, even after the introduction of the Nafion ionomer. Although Figure 4d indicates variations in the mesoporous region following ionomer coating, these are believed to be fitting artifacts and are not substantial enough to significantly alter the overall structure of the mesopores, which remain predominantly accessible. This contrasts with other catalysts where ionomer introduction has resulted in notable pore blockage, as identified by the lost peaks observed in pore size distributions.

We trust this explanation clarifies both the subtle shifts observed in our pore size distributions and the rationale behind the illustrations used in our figures.

Action taken:

Revision in Manuscript:

Subtle variations in the pore width within the mesoporous region were also observed in both Pt/HOMC and Pt/Vulcan following the introduction of ionomer. Specifically, Pt/Vulcan experienced an increase in pore size from 2.0 nm to 2.5 nm, and Pt/HOMC from 4.0 nm to 4.8 nm. These changes are attributed to modifications in adsorbate-adsorbent interactions. For Pt/HOMC, the C-BET values for the uncoated and ionomer-coated catalysts changed from 21.03 to 3.98, respectively. The significant reduction in C-BET value with ionomer coating suggests weakened interaction dynamics between nitrogen and the catalyst surface, necessitating higher partial pressures to fill the same pores. This adjustment results in a rightward shift in the adsorption isotherms, and consequently, an apparent increase in the estimated pore sizes. These observations align with findings from studies using ionic liquids and silica⁴³. Despite these alterations, the mesopore openings in Pt/HOMC remain largely intact, indicating that the ionomer segments neither fill nor block the mesopore openings.

8. And, if no sulfonate adsorption at low potential range on Pt/Ketjenblack and Pt/HOMC, why the diminution in the white line intensity is inconsistent? And what is the reason for the diminution in the white line intensity at 0.6 V?

Response:

Thank you for your insightful queries regarding the observations of diminished white line intensity at low potential ranges in Pt/Ketjenblack and Pt/HOMC. Several studies have similarly noted a diminution in white line intensity at around 0.6 V. For instance, references can be found

in published works (e.g., DOI: 10.1021/acsaem.0c02326, DOI: 10.1039/C0CP01434F). In these cases, the reduction in intensity has been considered negligible and has not been interpreted as indicative of any significant changes in the electronic state of platinum.

This consistent observation across various studies suggests that it is a real electronic effect rather than an experimental error. However, it is negligible and any assumption that this reduction indicates a change in the Pt electronic state would likely be an overestimation. Notably, significant changes at this potential are generally not expected unless there is a significant increase in white line intensity, as observed in Pt/Vulcan, which indicates the adsorption of ionomer. Consequently, this observed phenomenon is not typically associated with substantive electronic or structural modifications within the platinum catalysts.

Action taken:

Revision in Manuscript:

For Pt/Vulcan XC-72, a significant increase in white line intensity at 0.6 V indicates the adsorption of sulfonate, reflecting the close contact of Pt nanoparticles with ionomers. In contrast, Pt/Ketjenblack and Pt/HOMC displayed a slight diminution in white line intensity at the same potential. This diminution is a common observation, as noted in several studies.⁴⁴⁴⁸ In these studies, such reductions in intensity are generally considered negligible and not indicative of significant changes in the platinum's electronic state. This suggests that while the diminution is observable, it should be interpreted within the inherent fluctuations typical in in situ XANES measurements. Hence, any assumption that this slight reduction indicates a substantive change in the Pt electronic state would likely be an overestimation. Following this diminution, Pt/Ketjenblack and Pt/HOMC experienced an increase in white line intensity from 0.7 V onwards, attributed to *OH adsorption. This pattern suggests that at lower potentials, no sulfonate adsorption occurs, supporting the hypothesis that most Pt nanoparticles in these catalysts, located inside the mesopores or at their mouths, are not in direct contact with the ionomers on the outer surface of the carbon support.

9. For the section of conclusion, author briefly summarize the results of deposited platinum on diverse carbon support for the influence of ionomer coverage. However, it is just the assumption and explanation based on the results. Authors should give a more helpful viewpoint for the reason why the Pt nanoparticles can locate at the mesopore mouths, which is significant for the fabrication of highly active electrocatalyst.

Response:

Thank you for emphasizing the importance of understanding why Pt nanoparticles preferentially locate at the mesopore mouths. We agree that elucidating this phenomenon is crucial for advancing the design of highly active electrocatalysts.

In our study, we have chosen Pt/HOMC as a model system to specifically explore the interaction between the platinum nanoparticles and the mesoporous carbon support. The 1 wt% Pt loading was carefully designed in accordance with the mesoporous channel capacity of the support, as detailed in the Supplementary Information. This precise control allows us to systematically study the interaction dynamics under well-defined conditions.

On mesoporous supports, metal nanoparticles typically form within the mesoporous channels rather than on the outer surface, a phenomenon consistent with observations in similar systems (e.g., DOI: 10.1039/C0CC04719H, 10.1021/cs400506q, 10.1021/acscatal.5b02371). For our

system, the relatively short impregnation time (30 minutes) and the higher viscosity of ethylene glycol compared to water contribute to a shorter diffusion depth of Pt precursors into the mesoporous channels. Moreover, the polyol method allows for pH control that finely tunes the resulting Pt nanoparticle size to match the mesopore openings. These factors, combined with the intentionally designed low Pt loading, facilitate the formation of Pt nanoparticles at shallow depths at the mouths of each mesoporous channel. Theoretically, these sites are energetically favourable as the nanoparticles stabilize at lower energy sites to reduce the system's overall energy.

Although a detailed investigation into the energy landscape and the precise mechanisms driving the preferential localization of Pt nanoparticles at mesopore mouths is beyond the scope of this particular study, we believe our findings lay a foundational understanding that could guide future research in this area.

To address this complex topic adequately, we envisage future studies specifically designed to probe the energy profiles and material interactions at the mesopore mouths, utilizing advanced characterization techniques that can provide deeper insights into these phenomena.

Action taken:

Revision in Manuscript:

This study highlights the critical role of mesoporosity in designing carbon supports for ORR catalysts and suggests avenues for further research. The superior performance of the Pt/HOMC is achieved via preferential deposition of Pt NPs at the mesopore openings, a phenomenon driven by the minimization of surface energy. This specific positioning is facilitated by the relatively short impregnation time (30 minutes) and the high viscosity of ethylene glycol, which limits the diffusion depth of Pt precursors into the mesoporous channels. Additionally, the polyol method allows for pH control, finely tuning the resulting Pt nanoparticle size to match the mesopore openings. These factors, combined with the intentionally designed Pt loading, ensure that Pt nanoparticles form at shallow depths at the mouths of each mesoporous channel, optimizing accessibility and reactivity. To transition this model catalyst to practical use, development of carbon supports with a higher amount of correctly sized mesopores—to hold more Pt nanoparticles at the pore mouths—is essential. Future studies should focus on translating these findings into practical catalyst designs with higher Pt loadings. Such efforts aim to optimize the balance between catalytic performance and material costs, enhancing the viability of these catalysts in real-world applications.

Reviewer #2 (Remarks to the Author):

In this manuscript, Wang et al demonstrate how carbon support microstructure can control the properties of a fuel cell catalyst, and furthermore control ionomer structure. This is an important topic relevant to many types of electrocatalysts, and it is a famously difficult field of experimentation. Catalyst model systems using three different types of carefully selected carbon support were fabricated, which demonstrate significant changes in gas sorption properties. A new variant of operando X-ray spectroscopy cell is described and benchmarked, and an operando XANES measurement is shown, although the connection between the X-ray and gas sorption work are not very strong. This manuscript has a few substantial issues, which if corrected would make this a nice study of short length suitable for publication in a high impact journal.

1. In the introduction, it is fair to be a bit harsher on the various electrochemical methods for probing ionomer interactions. While calling them 'quantitative' is technically accurate, I just wish we knew what we are quantifying! There is virtually no structural evidence for the necessary assumptions. For example, the "dry" CO stripping measurement likely responds to more than just the Pt in contact with ionomer, for a variable definition of "in contact". That said, I was quite surprised not to see the "CO displacement" experiment run in this work. Unlike the CO stripping, you polarize the cell and then flow CO in, generating a transient that allows the charge of desorbing species at the interface to be integrated. This technique is now rather popular, and is capable, in theory, of calculating the ionomer coverage on Pt catalyst surfaces under relevant conditions.

Response:

Thanks for the suggestions and we agree that the CO displacement under MEA testing conditions would be very beneficial in determining the ionomer coverage on Pt catalysts. However, in the GDE half-cell setup employed in this project, the existence of liquid electrolyte would cause difficulty in interpreting the CO displacement results due to the co-adsorption of electrolyte anions and cations (doi: 10.1149/2.0551702jes). Nevertheless, we recognize the importance of this technique and its growing popularity in the field. In light of your feedback, we plan to explore the possibility of incorporating CO displacement experiments in future studies, potentially using a different setup that mitigates the influence of co-adsorbed species.

Action taken:

Revision in Manuscript:

Another emerging technique, namely CO displacement, was employed to quantify the adsorption of perfluorosulfonic acid ionomer by comparing the displacement charge and CO stripping charge. However, it again requires MEA setup to avoid the existence of liquid electrolyte, which would cause difficulty in interpreting the CO displacement results due to the co-adsorption of electrolyte anions and cations.³⁴ In contrast, X-ray absorption spectroscopy (XAS) could probe the changes caused by the adsorption of sulfonate ending group from the NafionTM ionomer in operando with a GED setup.

2. The choice of extremely low 1wt% Pt loading is interesting in the sense of building a model system, but limits the extension of this model to practical catalysts where 30-70 wt% Pt are routinely employed. I admit I did not really understand what the authors were trying to do here the first time I looked through the paper, and was ready to criticize this unrealistic feature. Part of my comprehension issue arised from the concept/objective of this work not being introduced earlier in the introduction, and only shows up mid way through the results and discussion section, where it is not connected to the literature review. One question of these low loadings is whether the polyol method selectively deposits Pt at specific points on the carbon support. These high energy adsorption sites may strongly influence the pore structure and nature of the blocking, which is unique to the author's system.

Response:

We appreciate the reviewer's observation regarding the choice of an extremely low 1wt% Pt loading, which indeed serves as a foundational element of our model system. Our primary

objective with this configuration was to specifically explore the impact of carbon support porosity on catalyst behaviours. This strategy was chosen to simplify the system and isolate specific phenomena, thus avoiding the complexities introduced by higher metal loadings. Importantly, the polyol method does selectively deposit Pt nanoparticles at designated sites on the three different carbon supports, providing a set of distinct model catalysts for our study.

In response to your feedback, we have emphasized that we are using model catalysts in introduction. Furthermore, we acknowledge that translating this model to practical, high-load Pt electrocatalysts presents certain limitations. To address these concerns, we have included some perspective for future work in the conclusion, highlighting the potential gaps while translating to high Pt loadings as seen in practical catalyst applications.

Action taken:

Revision in Introduction in Manuscript:

Herein, we prepared sustainable highly ordered mesoporous carbon (HOMC) from xylose and used it to rationalize the effect of carbon support morphology and porosity on Pt-ionomer interactions with a series of model Pt/C catalysts, via the use of advanced characterisation techniques.

Revision in Conclusion in Manuscript:

This study highlights the critical role of mesoporosity in designing carbon supports for ORR catalysts and suggests avenues for further research. The superior performance of the Pt/HOMC is achieved via preferential deposition of Pt NPs at the mesopore openings, a phenomenon driven by the minimization of surface energy. This specific positioning is facilitated by the relatively short impregnation time (30 minutes) and the high viscosity of ethylene glycol, which limits the diffusion depth of Pt precursors into the mesoporous channels. Additionally, the polyol method allows for pH control, finely tuning the resulting Pt nanoparticle size to match the mesopore openings. These factors, combined with the intentionally designed Pt loading, ensure that Pt nanoparticles form at shallow depths at the mouths of each mesoporous channel, optimizing accessibility and reactivity. To transition this model catalyst to practical use, development of carbon supports with a higher amount of correctly sized mesopores—to hold more Pt nanoparticles at the pore mouths—is essential. Future studies should focus on translating these findings into practical catalyst designs with higher Pt loadings. Such efforts aim to optimize the balance between catalytic performance and material costs, enhancing the viability of these catalysts in real-world applications.

3. The gas sorption measurements are a little puzzling to me. Better analysis of these curves could be quite informative towards understanding the microstructure of the HOMC, which is at the heart of this manuscript. No doubt the HOMC carbon has decent surface area and some porosity, visible in the TEM. But for a paper that relies very strongly on a conceptual model like the cartoons in Fig 2, the question becomes how correct these are.

Firstly, I don't see the traditional capillary condensation and hysteresis for HOMC I would normally expect for open mesopores. My understanding was that closed or severely bottlenecked pores produce less, and more complex hysteresis, while open cylindrical pores like those drawn in Figs 1 and 2 should give a very pronounced hysteresis with a classically shaped isotherm. When I look at the sorption data in Fig 1 of the manuscript, the curves are not labelled, but I assume the one starting at low P/Po is the sorption curve. The desorption curve actually crosses back underneath the sorption curve, which I would like a physical explanation for. The common errors

I have seen, like setting too short of an equilibration time, usually result in undershooting the sorption curve, and increasing hysteresis, not the inverse. Usually I see DFT methods used to calculate the pore size distribution from the desorption half of the hysteresis loop, although adsorption methods can also be used. I did not understand which parts of which curves were included in this model. The P/P_0 log plot was not included in the SI, so I could not review the low pressure data. Confusingly the adsorption data of the bare HOMC in Figure 1 does not match the adsorption/desorption data in Figure 2a, even though as far as I can tell, this is the same material. In contrast to Figure 1, the data in Figure 2 appears to be exceptionally high quality. For example, Fig 2b shows the H3 style isotherm characteristic of wedge-shaped pores in Ketjen Black which is absolutely textbook, and in line with the cartoon.

Response:

We appreciate the detailed feedback regarding the gas sorption measurements and the interpretation of the isotherm data. Below, we address each point raised:

Capillary Condensation and Hysteresis: We agree that traditional capillary condensation and hysteresis are often expected for mesopores due to the capillary condensation phenomenon influenced by pore size. According to the IUPAC report on physisorption of gases (<https://doi.org/10.1515/pac-2014-1117>), the critical pore diameter for nitrogen adsorption@77K at which capillary condensation typically begins to exhibit pronounced hysteresis is around 4 nm. Pores smaller than this critical size generally do not show significant hysteresis due to the lack of capillary condensation as shown in below figure.

The phase diagram of adsorption hysteresis criticality for nanoporous materials. doi.org/10.1016/j.molcata.2005.05.013

Our measurements indicate that the cylindrical pores in our HOMC material largely fall within the 3-4 nm range, closely approaching this critical threshold. This dimension is significant because it places many of the pores at a point where the transition from exhibiting to not exhibiting hysteresis might occur, which could explain the missing hysteresis observed in our isotherms.

Additionally, the shape and connectivity of the pores significantly impact the hysteresis. For example, ink-bottle shaped pores (narrow necks with wider bodies) can cause delayed

evaporation during desorption, resulting in a more pronounced hysteresis loop, which is the case for Ketjenblack. In contrast, our HOMC materials feature highly uniform cylindrical mesopores with very rough walls (caused by the soft templating and proved with the CO₂ sorption analysis) and open ends on both sides. These factors minimize the differences observed during adsorption and desorption, contributing to the absence of a pronounced hysteresis in our measurements. Even with the absence of the hysteresis, we are confident in claiming the successful development of mesoporous channels based on the well studied mechanism of block copolymer self-assembly and the TEM images showing clear mesopores agreeing with the results given by the pore size distribution.

Pore Size Distribution Calculation: In our original manuscript, we utilized the built-in pore size distribution model based on nonlinear DFT provided by Micromeritics. Unlike the conventional BJH model which allows one to choose between adsorption or desorption data, NLDFT approach considers information from both the adsorption and desorption branches (<https://doi.org/10.1515/pac-2014-1117>). This dual consideration allows for a more reliable assessment of the pore size distribution, especially in materials exhibiting hysteresis behaviours such as the Ketjenblack materials examined in our work.

Low Pressure Data: We have provided the P/P₀ log plot below as well as in the supplementary information to allow for a comprehensive review of the low-pressure data, ensuring that all data points used in our analysis are transparent and accessible.

Action taken:

Added figure in SI:

Figure S12. N₂ sorption data plotted with log(p/p^0) scale.

Inconsistent bare HOMC data and crossback desorption curve in Figures 1: We apologize for the oversight in not addressing the inconsistencies between the data presented in Figures 1 and 2. During the initial phase of this project, gas sorption measurements were conducted using the Tristar Plus II instrument from Micromeritics, which provided the data shown in Figure 1. As the project progressed, we transitioned to the more advanced 3Flex instrument from Micromeritics, known for its superior control and resolution of pressure measurements. Consequently, the data presented in Figures 2 and 4 exhibit higher quality and resolution. To ensure consistency and accuracy, we have repeated the gas sorption measurements for bare HOMC using the 3Flex instrument. The new measurements are consistent with the high-quality data shown in Figure 2 and address the issue of the crossback desorption data, which is likely caused by inefficient equilibrium and is a measurement artifact we should have addressed initially. Accordingly, we have updated Figure 1 with the new data obtained using the 3Flex instrument. We appreciate the reviewer's attention to this detail, which has helped us improve the clarity and reliability of our manuscript.

Repeated N₂ sorption analysis of bare HOMC on 3flex instrument.

Action taken:

Updated Figure 1 in Manuscript:

Figure 1. a. Preparation of highly ordered mesoporous carbon (HOMC) via sulfuric acid mediated hydrothermal carbonisation of xylose. b. TEM image of HOMC. c. SEM image of HOMC d. N₂ sorption isotherms for HOMC at 77K. e. NLDFT pore size distribution for HOMC.

5. Creative control experiments may significantly enhance the credibility of the cartoon model. N₂ isotherms performed at 77K are extremely sensitive to pore blocking. Other gases at higher temperature are usually employed to evaluate whether pores are actually capped, and the difference even with an argon isotherm at 87K can be dramatic. Since PEMFCs do not operate at 77K, there is some thought needed to the validity of this aspect of the model. More precise language may also help a reader here. For example, after Pt deposition it is claimed that nearly 90% of the HOMC porosity is lost. However, this porosity still exists, and is merely inaccessible under the measurement conditions, which is mentioned later on in the discussion. Grinding or lightly milling the Pt/HOMC could break the support particles into pieces, reopening the sealed pores and restoring the enormous original surface area.

Response:

We acknowledge that N₂ isotherms performed at 77K can be sensitive to pore blocking, and that evaluating the pore structure with other gases at higher temperatures can provide additional insights. In response to this, we have conducted CO₂ sorption experiments for all our samples. CO₂ sorption, performed at 273K, is less susceptible to pore blocking effects and provides a more accurate representation of the pore accessibility under conditions closer to those in PEMFC operations. The porosity under 1 nm is better probed with CO₂, agreeing very well with the observations done by the N₂ sorption upon analysis:

1. The introduction of Pt sites has reduced the microporosity slightly for all the three samples as some of the micropores under 1 nm become no longer measurable with the hinderance of Pt nanoparticles. It is worth noting that the introduction of Pt onto HOMC resulted in much less effect when probed using CO₂ compared to N₂. This supported our analysis that the loss of accessible porosity under N₂ sorption is caused by the Pt location on the mesoporous openings, which is beyond the detection range of CO₂ sorption.
2. The introduction of ionomer gave much more significant effect on blocking the micropores as suggested in the original manuscript.

We have included the CO₂ sorption results in the Supplementary Information and relevant discussion in the revised Manuscript to provide a comprehensive analysis of the pore structure. We also appreciate the suggestion for more precise language. In the revised manuscript, we have clarified that the reported 90% loss of porosity after Pt deposition refers to the inaccessibility of the porosity under the specific measurement conditions of N₂ sorption at 77K due to the location of the Pt nanoparticles, rather than a complete loss of porosity.

Action taken:

Revision in Manuscript:

For Pt/HOMC, the introduction of ionomer predominantly affected the microporosity, as evidenced by comparisons between Figures 4a and 4d, indicating that ionomer segments covered the carbon surface and blocked the micropores. Similarly, in Pt/Vulcan, the microporous structure was almost entirely obstructed due to ionomer coverage, as depicted in Figures 4c and 4f. In contrast, Pt/Ketjenblack exhibited a significant alteration in porosity after the incorporation of ionomer, leading to a pronounced decrease in both micropores and mesopores (Figures 4b and 4e). The blocking of micropores on the three supports were further confirmed by CO₂ sorption results (Figure S14).

Added text in Methods in Manuscript:

Carbon dioxide isothermal adsorption/desorption was conducted on a Micromeritics TriStar II Plus system at 273 K following the same degassing conditions as N₂ sorption. The pore size distribution was calculated using slit geometry and DFT model (CO₂@273 on Carbon) in-built with TrisStar II Plus software.

Added figure in SI:

Figure S14. CO₂ sorption isotherms and pore width distributions.

6. That Yargaladda paper discussed in the introduction is frustrating, since it does not show any data like TEM or isotherms or electrochemistry. It also does not say how the catalyst films are actually made or deposited, the parameters of which play a massive role in the performance of the device. That paper, and its very high number of citations is a good example of the type of sparsely characterized work being done in the field, which the present authors are able to surpass using high quality data.

Response:

We thank the reviewer for highlighting the potential limitations in the Yargaladda study discussed in our introduction. We acknowledge that the lack of detailed experimental data in

Yarlagadda et al.'s work could indeed impact the interpretability and applicability of their findings.

In citing this work, our intention was to provide a contextual framework that outlines the state of research within the field, particularly focusing on the recognized influence of carbon support structures on ionomer distribution. However, we appreciate that our manuscript could benefit from a clearer exposition of these issues to prevent any misunderstanding about the reliability of the cited study as a foundation for our research.

In response to your valuable feedback, we have revised our manuscript to include a critical evaluation of the previously cited study. We now explicitly highlight the limitations of Yarlagadda et al.'s methodology and the absence of essential characterization details, which could undermine the robustness of their conclusions. This amendment aims to clarify our intent and enhance the scientific rigor of our study.

We believe these revisions will make the objectives and contributions of our research clearer and hope that the updated manuscript now adequately addresses the concerns raised.

Action taken:

Revision in Manuscript:

A pivotal conclusion from their research was the assertion that mesopores between 4 to 7 nm are the most accessible, and thus, could enhance the ORR performance. This conclusion was drawn based on the outlined 4-7nm pore volume in their statistical analysis of the carbon support structure. While their work is foundational and offers intuitive insights, it's essential to note that the nanostructure of catalyst supports is intricate. The emphasis on the 4-7nm pore volume, though significant, might not capture the entire picture. **While their study has been influential in identifying key relationships between carbon support nanostructure and ionomer distribution, it is essential to address certain shortcomings in their experimental methodology and absence of essential characterisations, which could undermine the robustness of their conclusions.**

7. The CVs in Figure 3 do not show any H-UPD phenomena, or even clear Pt oxidation, which implies they are very strongly contaminated even after one sweep to high potential. High quality, clean CVs are the essential component for evaluating ORR activity and ECSA of Pt based PEMFC electrocatalysts. While the high Pt to carbon ratio will make the capacitance a bit larger than usual, we can estimate the size of these Pt specific features from the CO stripping peaks, and see they are absent. This is quite a show-stopper, and I cannot recommend publication without it, unless comparisons of catalytic activity and ECSA are removed. I recommend the authors activate their catalysts by cycling 0.05-1.2V at perhaps 100mvps for maybe 25 cycles or until stable, to clean the surface before attempting any sort of CO sorption or ORR. To obtain the ECSA with CO stripping, it is best to subtract the subsequent, CO-free CV from the CO-bound one, instead of trying to use a linear baseline correction. Of course there is always discussion of what activation does to catalyst structure, but without understanding what state the Pt is in, going the next step to probing the Pt-ionomer interactions seems impossible.

Response:

Thank you for the suggestion and we fully agree on the necessity of clean CV curves for accurate analysis. We have developed and employed a comprehensive experimental protocol when using the gas diffusion half cell and detailed the protocol as below:

Step	Method	Parameter	
1 Determination of iR drop	EIS at OCP	Gas flow	N ₂ (~200 mL/min)
		EIS frequency range	10000-2 Hz
		EIS amplitude	0.01 V _{RMS}
		Poential	OCP
2 Electrochemical cycling	CV	Gas flow	N ₂ (~200 mL/min)
		Potential limits	0.05-1.05 V
		Scan rate	100 mV/s
		Number of cycles	~35 (until CV constant)
3 ORR (O ₂)	Potentiostatic staircase	Gas flow	O ₂ (~200 mL/min)
		Pt-reduction step	0.1V, 5min
		ORR potential range	1.0 -0.0 V
		Step size	50 mV
		Holding time	15s
4 ORR (O ₂)	LSV	Gas flow	O ₂ (~200 mL/min)
		Pt-reduction step	0.1V, 5min
		ORR potential range	0.0-1.0 V
		Scan rate	10 mV/s
5 Formation of CO-monolayer	Chronoamperometry	Gas flow	CO (~200 mL/min, 5 min), then N ₂ (~200 mL/min, 15min)
6 ECSA (CO)*	CV	Potential	0.1 V
		Gas flow	N ₂ (~200 mL/min)
		Potential limits	0.1-1.0 V
		Scan rate	20 mV/s
		Number of cycles	3

We have included 35 “cleaning cycles” of the catalysts at a sweeping range of 0.05 to 1.05 V and a sweep rate of 100 mV/s, prior to any measurements, ensuring the CV curve is stabilised after the cleaning step. A sample CV recording can be found below and even after 35 cycles and CV curve is stabilised, there is no noticeable peak for HUPD and Pt oxidation.

Electrochemical cleaning of the electrode with 35 cycles of CV.

Other than the electrochemical cleaning of each electrode, we have also implemented a cleaning protocol of the electrochemical cell being used. This includes an overnight soak of the

cell in a concentrated sulfuric acid bath followed by several hours of boiling in deionized water, ensuring any contaminants are thoroughly removed.

Despite these rigorous cleaning protocols for both the catalyst and the cell, H-UPD peaks were not observed in our CVs. We believe this phenomenon is not due to residual contamination but rather attributed to the substantial relative capacitance of the carbon support, which obscured the expected current signatures of H-UPD. To illustrate, we compared our in-house developed Pt/Vulcan, which has a Pt loading of 0.78%, with the commercial catalyst HiSPEC4000, featuring a Pt loading of 38%-41% on the same Vulcan XC-72 carbon support. The ratio of Pt ECSA to carbon surface area is approximately 0.0015 for our Pt/Vulcan (calculated as $41 \times 0.78\% / 210$) and 0.095 for HiSPEC4000 (calculated as $50 \times 40\% / 210$). This represents a 64-fold difference, rendering the H-UPD peak in our model catalyst systems unobservable due to the dominance of carbon capacitance.

Moreover, we can confirm that in our experiments, we used the same method for determining the ECSA. Specifically, we subtracted the subsequent CO-free cyclic voltammogram (CV) from the CO-bound CV to correct for any residual currents, instead of using a linear baseline correction. This approach ensured that we obtained a clear and accurate representation of the Pt surface, essential for our analyses on the Pt-ionomer interactions.

8. Unfortunately the same limitations about not knowing what the surface chemistry is like also applies to the XANES experiment, although the standards for operando cells are much lower. Increasingly high potentials will slowly clean the surface over time. In addition, we can't discount the influence of the X-ray beam, which generates considerable quantities of radicals and peroxides which clean loosely bound organic residues, and ionomer sidechain degradation products, which are powerful poisons known to bind very tightly to the Pt surface. Was any check made on the reversibility of the XANES spectra, like stepping back to 0.5V after the spectra at 1.0V was collected?

Response:

Thank you for your thoughtful inquiry regarding the experimental methodology used in our XANES experiments, particularly concerning the reversibility of the spectra. We indeed observed beam damage to the sample in the initial configuration. In order to help mitigate this, 120 microns of pyrolytic carbon attenuation was used to reduce flux on the sample and measurements for each potential were measured at several points along the pellet surface. While we did not systematically return to 0.5 V from 1.0 V across all data points, we incorporated a method where we stepped back by 0.1 V at random intervals to verify the consistency and reliability of the data. These step-back measurements combined with scanning across the pellet and the added attenuation confirmed that the spectra remained consistent and reproducible, reinforcing the integrity of our findings.

To provide a comprehensive view of our methodology and the additional checks performed during the experiments, we have included these details in the methodology section. This addition ensures transparency and allows for a more in-depth understanding of how potential fluctuations or hysteresis were managed and their impact assessed, addressing concerns about potential changes in surface chemistry over time and under different experimental conditions.

We hope this clarification addresses your concerns and illustrates our commitment to rigorous experimental procedures and data validation.

Action taken:

Added text in Manuscript:

After CV cleaning cycles of the catalyst, the spectra were collected at the open circuit potential before the potential was stepped up from 0.5 V to 1.0 V with 0.1 V intervals. For each potential, 5 XANES were collected. We also incorporated a method where we stepped back by 0.1 V at random intervals to verify the consistency and reliability of the data. This would also check if there were any hysteresis due to the irreversible damage or change to catalysts caused by the beam. These step-back measurements confirmed that the spectra remained consistent and reproducible, and these data were combined with the ordinary repeats for the merged spectra (see example in Figure S18).

Added figure in SI:

Figure S18. Pt L₃ edge absorption data collected for Pt/HOMC at 0.7 V, including 5 repeats collected after stepping up from 0.6 V and 1 repeat collected after stepping back from 0.8 V.

9. Was a spectra collected under CO atmosphere, or was this too unsafe?

Response:

Thank you for your question. Unfortunately, due to safety considerations, we were unable to conduct CO stripping measurements at the synchrotrons where we conducted our experiments.

10. White line intensity is an extremely flexible parameter, in that various authors routinely invoke small shifts to be correlated with any convenient structural change du jour, such as the strength of particle-support interaction, which also changes with potential. I was surprised not to see the particle support interaction mentioned at all, given the comprehensive XAS work on all the catalysts. I do quite like the XANES idea, but additional discussion re: the expected change in intensity would be helpful in supporting the interpretation, since so many different things could be happening.

Response:

Thank you for your insightful comments regarding the flexibility of interpreting white line intensity changes in XANES measurements and the importance of considering particle-support interactions.

While it is true that metallic nanoparticles can experience electronic interactions from the support, these interactions typically extend over only about one atomic distance (10.1103/PhysRevLett.76.3168, 10.1073/pnas.0505436102). Therefore, the majority of the Pt nanoparticles in our study would not significantly experience such interactions, which is why we neglect particle-support interactions in our analysis.

Regarding the expected changes in white line intensity, we have clarified in our manuscript that the adsorption of either ionomer or OH groups can cause an increase in intensity, though these occur at distinct potentials. We have endeavoured to make this distinction clear, supporting our interpretations with comprehensive data.

Action taken:**Revision in Manuscript:**

To further evidence the interaction between the ionomers and Pt NPs under ORR conditions, *operando* XAS analysis was carried out to investigate the specific adsorption of the ionomer under different potential over the three Pt/C. Although white line intensity change could potentially also be correlated with effect such as change in nanoparticle-support interactions, these interactions typically extend over only about one atomic distance for metallic nanoparticles.⁴⁴⁴⁵ Therefore, the majority of the Pt nanoparticles in our study would not significantly experience such interactions. On the other hand, induced by the adsorption of anions under potential, the electronic structure of Pt atoms changes accordingly. The more anions adsorbed onto the Pt surface the higher white line intensity would be expected due to the formation of Pt-O bond.⁴⁶

11. Is there a specific reason that entire spectra were collected? Why not just measure 1 point pre and post edge for normalization, and one at the max intensity? This would presumably allow for much faster data acquisition, and even operando XANES during cyclic voltammetry would be possible.

Response:

Thank you for your suggestion regarding the data acquisition technique for our *operando* XANES measurements.

We opted for a comprehensive spectral collection based on a well-established method detailed in doi: 10.1021/j150655a029. This method involves the quantification of d-band vacancies in Pt catalysts by systematically analysing both L2 and L3 edge absorptions. It requires meticulous alignment, background removal, and normalization across both edges, ensuring consistency and providing quantitative insights. This rigorous approach was previously validated in another study (10.1021/acsaem.0c02326) on ionomer-catalyst interactions, where it demonstrated strong correlations and robust results.

Given the complex interactions among porosity, ionomer, and catalyst in our system, this proven methodology was deemed the most straightforward for us to achieve the necessary precision and reliability in our observations.

Your suggestion to streamline the measurement process by focusing on key points for normalization and maximum intensity is intriguing and could indeed facilitate faster data acquisition. We appreciate this innovative approach and plan to explore its feasibility and potential benefits in future experiments.

Action taken:

Added text in Manuscript:

A quantification method⁴⁷ (detailed in Methods section) was adopted to consider the near edge absorption differences against Pt foil reference at both L₂ and L₃ edges. This method involves the quantification of d-band vacancies in Pt catalysts by systematically analysing both L₂ and L₃ edge absorptions, with meticulous alignment, background removal and normalization across both edges, ensuring consistency and providing quantitative insights.

12. The benchmarking of the half cell was obviously performed with a high degree of care, and the detailed descriptions and controls, eg checking for steady state conditions, is greatly appreciated.

Response:

Thank you very much for your positive feedback regarding the benchmarking of the half cell.

13. It would seem that electron tomography (10.1017/S1431927617011114) or FESEM (10.1021/acsaem.2c01790) are much better than regular TEM to understand whether catalysts are buried inside the carbon, where the 2D projected image provides very limited depth information.

Response:

Thank you for the suggestions, and we agree with the limited depth information provided by the 2D projected TEM images. However, due to the low loading (<1 wt% Pt) and the small dimensions of the platinum nanoparticles (~2 nm), the resolution offered by STEM or FESEM is limited in capturing the presence of Pt NPs.

Nevertheless, we have performed FESEM imaging using the instrument with the best resolution available to us (LEO Gemini 1525 FEGSEM). As evidenced by the images, we could spot some of the Pt NPs on the surface of the HOMC despite the low contrast, identified by the brighter little dots observed. However, for carbon Vulcan and Ketjenblack, the small size of the carbon particles increases the contribution of edge effects, making the contrast of the tiny Pt NPs extremely difficult to confidently identify. Despite these challenges, the FESEM images provide useful information regarding the distribution of Pt NPs on the three different types of carbon confirming there is no aggregations of the Pt NPs, supporting our analysis.

Pt/HOMC

100 nm*

EHT = 3.00 kV
WD = 2.5 mm

Signal A = InLens
Mag = 82.62 K X

Date :1 May 2024
Time :15:20:16

Pt/VC

Pt/KB

14. There are a few spots where the typo Kentjen is used instead of Ketjen.

Response:

Thank you for pointing out the typographical errors. We have carefully reviewed the manuscript and corrected all instances where "Kentjen" was mistakenly used instead of "Ketjen." We appreciate your attention to detail.

Reviewer #3 (Remarks to the Author):

The investigation into the effects of carbon support pore size on particle dispersion, catalyst-ionomer interaction, and cell performance has garnered significant attention. As outlined in the introductory section, the elusive nature of these theories is often attributed to the tortuous pore structure of current carbon supports and the limitations of characterization techniques. Wang et al. addressed this challenge by employing three distinct carbon supports with varying pore distributions to examine the correlation between pore size and catalyst-ionomer interaction. This study utilized operando X-ray absorption spectroscopy and gas sorption analysis techniques. Despite the advanced nature of the XAS method, the results cannot distinctly differentiate among the three types of support because of the inconsistent intrinsic activities of Pt particle and yielding no novel conclusions. The following issues need addressing before further consideration:

Response:

Thank you for providing us with the opportunity to improve our work. We address your detailed comments in the following responses.

1. The manuscript contains typos, such as the TEM comparison in Figure S3 instead of Figure S1 (line 172).

Response:

Thank you for bringing this error to our attention. We have corrected the reference to Figure S3; it should indeed be Figure S1 as mentioned on line 172. We have further reviewed the manuscript thoroughly to ensure that all other typographical errors are identified and rectified to maintain the accuracy and clarity of our documentation. We appreciate your vigilance in helping us improve the quality of our manuscript.

2. Pt loading in the samples is extremely low (<1 wt%) based on ICP data, yet the Pt particle size is not small and is similar to commercial Pt/C. Clarification is needed on how this low loading with normal-sized Pt particles reduces HOMC support porosity by 90%.

Response:

Thank you for your comment. We agree that we should be more precise with the description of the lost porosity observed after introducing Pt nanoparticles onto the HOMC. The pores are still present in the Pt/HOMC but become inaccessible due to the location of the Pt at the pore openings. We have therefore revised our language to clarify that the reported 90% loss of porosity after Pt deposition refers to the inaccessibility of the mesoporosity under the specific measurement conditions of N₂ sorption at 77K, due to the location of the Pt nanoparticles, rather than a complete loss of porosity.

Action taken:

Revision in manuscript:

Interestingly from Figure 2a-f, one can observe that the porosity of both Ketjenblack and Vulcan XC-72 was largely retained after the Pt loading, **while 90% porosity of HOMC became inaccessible.**

Lastly, on HOMC, Pt NPs are deposited at the surface and located at the opening of the mesoporous channels, **making the mesopores no longer accessible for nitrogen sorption measurements** (Figure 2g).

3. The quality of TEM in Figure S3 is subpar. The figures should have the same magnification bar, as the current representation suggests a higher Pt loading for Pt/HOMC, with Pt particles mostly aggregating around the surface of the HOMC support.

Response:

We appreciate your observation regarding the quality and consistency of the TEM images in Figure S3. To address your concerns, we have now standardized the magnification scale bars across all TEM images for better comparison.

Additionally, we have provided supplementary TEM images for each sample to offer a more comprehensive representation of the Pt distribution.

We acknowledge that the visual impression of Pt loading appears higher for Pt/HOMC compared to other supports. This is due to the nature of TEM as a 2D projection technique. The HOMC carbon structure is significantly larger, measuring around 300-400 nm, while Vulcan and Ketjenblack are approximately 30-60 nm. This results in a 5-10 fold difference in the total depth projected per unit area in the images. Therefore, for an accurate measurement of Pt loading, we have employed ICP-MS analysis after complete digestion of the catalysts. The detailed raw data from this analysis is included in Figure S1.

Your observation that Pt is more concentrated on the surface of HOMC is consistent with our model, which suggests that Pt nanoparticles are primarily located in the mesoporous openings of the HOMC. This finding aligns with our gas sorption analysis and XPS results.

Action taken:

Revision in SI:

Figure S4. TEM images of Pt/HOMC, Pt/Vulcan XC-72 and Pt/Ketjenblack indicating similar particles sized obtained for 3 catalysts.

4. It is better to provide XRD data to estimate the particle size of the three catalyst types.

Response:

Thank you for your suggestion regarding the provision of XRD data to estimate the particle size of the three catalyst types. We agree that XRD is a crucial technique for characterizing

crystalline materials. However, in the series of samples prepared for this manuscript, which feature an extremely low loading of Pt (<1 wt%) serving as a model system to study Pt-ionomer interactions, the diffraction peaks specific to Pt were not detectable. We have included the XRD data in Figure S6 of our submission. As anticipated, the extremely low concentration of Pt crystalline phase, coupled with the predominantly amorphous nature of the carbon support, unfortunately masks the Pt peaks, rendering them unidentifiable within the XRD spectrum.

To accurately measure the particle size and overcome the limitations posed by XRD in this instance, we employed manual measurements using TEM images. The TEM analysis allows us to directly observe and measure the Pt nanoparticles dispersed on the carbon support, providing a reliable assessment of their size and distribution.

We believe that the TEM data, presented in our supplementary figures, effectively complements the XRD findings and offers a comprehensive understanding of the catalyst size. We hope this approach addresses your concerns.

Action taken:

Added text in Manuscript:

XRD of the three Pt/C catalysts were also collected (Figure S6). Due to the low loading of Pt, the diffraction peaks specific to Pt were not detectable.

Added figure in SI:

Figure S6. XRD patterns of the three catalysts.

5. ORR activity comparison should be provided, evaluated under the liquid cell using the Glassy carbon rotation disk electrode. Excluding ionomer or transport impacts, the intrinsic Pt activities of the three catalysts should be demonstrated as equal.

Response:

Thank you for your suggestion regarding the ORR activity comparison. We performed the evaluation under a liquid cell using a glassy carbon rotating disk electrode (RDE), specifically excluding ionomer to avoid its impact on mass transport. For each of the three catalysts, we used a consistent loading of 20 μ L ink of a 1 mg/mL catalyst suspension in a 1:1 mixture of ethanol and water. The results were included in Figure S7. Our measurements showed no

notable differences in the ORR activities among the three catalysts, confirming that they have equal intrinsic Pt activities.

Action taken:

Revision in Manuscript:

The three Pt/C catalysts showed no notable difference in the ORR activity under RDE setup (Figure S7), confirming that the Pt NPs have similar intrinsic activities. However, the ORR performance measurements done in an RDE configuration is limited by the low solubility of O₂ in electrolyte and thus would offer limited information on mass transport, which becomes critical at high currents. To investigate the effect of carbon support nanostructure, all ORR tests were therefore carried out in a GDE half-cell (Figure 3a) purchased from Gasketel with in-house modification.

Added figure in SI:

Figure S7. ORR polarization curves measured in RDE half-cell. Measurements were done in 0.1M HClO₄ electrolyte at room temperature with 200 mL/min oxygen flow and 20 mV/s cathodic scanning rate. Pt rod as the counter electrode and RHE as the reference electrode were used in the same electrolyte chamber without membrane separation, Potential was corrected for IR compensation post measurement.

6. An explanation is required for why Pt/Ketjenblack exhibits a much lower ECSA and poorer performance than Pt/Vulcan or Pt/HOMC in Figure 3, even at high potential regions. These data seem inconsistent with literature results, indicating potential issues with the Pt dispersion approach.

Response:

Thank you for your comment. We developed these three catalysts specifically to create a model system for studying ionomer-Pt interactions. To this end, we intentionally kept the Pt loading on these catalysts below 1 wt%, which is significantly lower than the typical loadings found in commercial Pt/C catalysts. Our characterization revealed that a considerable portion of the Pt on Ketjenblack was situated within inaccessible pores, leading to the low ECSA we observed. In other conventional Pt/Ketjenblack catalysts, this issue is somewhat mitigated by much higher Pt loadings, usually exceeding 20 wt%, which diminishes the effects of inaccessible Pt sites.

Regarding the poorer performance of Pt/Ketjenblack at higher potential regions, this is largely attributed to mass transport limitations and the accessibility of active sites. Compared to the

other two catalysts, Pt/Ketjenblack suffers significantly from reduced accessibility of active sites due to ionomer sealing and poisoning effects. It is also important to note that in the GDE setup, the 'high potential regions' are also largely affected by mass transport issues due to the much higher currents compared to those in RDE setups. This underscores the value of your previous advice on evaluating the catalysts under RDE conditions without ionomer. Observations under RDE reveal that the performance among the three catalysts shows negligible differences, suggesting comparable intrinsic activity of the Pt. We appreciate your attention on this, and hope this explanation addresses your concerns regarding the disparities with literature results.

Action taken:

Revision in Manuscript:

The three Pt/C catalysts showed no notable difference in the ORR activity under RDE setup (Figure S7), confirming that the Pt NPs have similar intrinsic activities.

The ECSA values, measured via CO stripping, were 41 m²/gPt for Pt/Vulcan, 33 m²/gPt for Pt/HOMC, and 24 m²/gPt for Pt/Ketjenblack, respectively. For Pt/HOMC, the slightly lower ECSA value is attributed to the unique positioning of the platinum nanoparticles. Predominantly located at the entrances of the mesopores, a significant match between the size of the Pt nanoparticles and the mesopore openings leads to a substantial portion of the surface atoms being obstructed by the mesoporous structure. Consequently, these atoms are rendered electrochemically inactive, impacting the ECSA. In contrast, the Vulcan XC-72 carbon support, primarily consisting of much smaller micropores (<1 nm) that cannot trap Pt nanoparticles, allows most Pt nanoparticles to be situated on the outer surface of the carbon support. This configuration results in a higher ECSA. This phenomenon is supported by the study³⁹, which suggests that metal nanoparticles located on the outer surface of carbon support expose more surface atoms compared to those trapped in porous structures. The ECSA of Pt/Ketjenblack is even lower due to its highly tortuous pore structure, which leads to a much higher amount of Pt nanoparticles being sealed within these pores, rendering them hardly accessible.

7. With the addition of the same amount of ionomer, why only the surface area of Pt/Ketjenblack experiences around a 73% decrease, as the ionomer film also constructed on Vulcan or HOMC supports?

Response:

Thank you for your inquiry regarding the specific behavior observed in the BET surface area of Pt/Ketjenblack upon the addition of the same amount of ionomer, compared to that on Vulcan or HOMC supports. As detailed in our manuscript, the considerable 73% reduction in surface area for Pt/Ketjenblack primarily stems from the unique characteristics of Ketjenblack, whose high surface area is largely contributed by the abundant micropores. Upon ionomer addition, which appears as segmented clusters/films on the surface of the carbon particles, significant sealing of the micropores occurs. This is particularly evident in the loss of surface area from the sealing of micropores below 2 nm, as indicated by the missing peak at 1 nm in Figure 4e. Additionally, the inherent bottleneck pore shape of Ketjenblack (evidenced by the hysteresis in the N₂ sorption isotherms) causes a substantial portion of mesopores, especially those with bottleneck sizes around 2-3 nm, to also become inaccessible due to ionomer coverage.

In contrast, the Pt/Vulcan and Pt/HOMC catalysts, which have different pore structures, do not exhibit the same extent of surface area reduction. On Pt/HOMC, the mesopores are uniformly around 4 nm in opening, which is too large to be fully covered by the ionomer segments. Meanwhile, Pt/Vulcan does not possess bottleneck mesopores that could be significantly affected by the sealing of surface micropores. The surface area loss of both HOMC and Vulcan solely came from the loss of their micropores on the outer carbon surface, giving a much smaller portion compared to Ketjenblack.

We hope this response has clarified the rationale behind the observed phenomena and appreciate the opportunity to enhance the understanding of our findings.

Action taken:

Revision in Manuscript:

N₂ sorption before and after the introduction of ionomer was measured to understand the ionomer coverage (Figure 4a-c & Figure S12, S13). For Pt/HOMC, the introduction of ionomer predominantly affected the microporosity, as evidenced by comparisons between Figures 4a and 4d, indicating that ionomer segments covered the carbon surface and blocked the micropores. Similarly, in Pt/Vulcan, the microporous structure was almost entirely obstructed due to ionomer coverage, as depicted in Figures 4c and 4f. In contrast, Pt/Ketjenblack exhibited a significant alteration in porosity after the incorporation of ionomer, leading to a pronounced decrease in both micropores and mesopores (Figures 4b and 4e). The blocking of micropores on the three supports were further confirmed by CO₂ sorption results (Figure S14).

8. The rationale for the increase in pore width of ionomer-coated Pt/Vulcan and Pt/HOMC compared to the uncoated situation needs to be elucidated.

Response:

We appreciate the reviewer's attention to the detailed observations regarding pore width variations in ionomer-coated Pt/Vulcan and Pt/HOMC catalysts compared to their uncoated counterparts. However, we would like to highlight the shift in the peak position is relatively small, with Pt/Vulcan displaying an increase from 2.0 nm to 2.5 nm, and Pt/HOMC from 4.0 nm to 4.8 nm. This change suggests subtle modifications in the pore structure upon ionomer coating.

The alteration in pore width is attributed to changes in adsorbate-adsorbent interactions. For instance, in the case of Pt/HOMC, the C-BET values for the uncoated and ionomer-coated catalysts are 21.03 and 3.98, respectively. The decrease in the C-BET value with ionomer coating suggests altered interaction dynamics between nitrogen and the catalyst surface, which contributed to the observed shift in the nitrogen adsorption isotherms, and subsequently, to the estimated pore sizes. With a reduced C-BET value (weakened interaction), it would need higher partial pressures to fill the same pores leading the curve to shift right and the results to be artificially shifted to higher pore sizes.

The relevance of this finding is supported by analogous phenomena reported in systems utilizing ionic liquids and silica, as detailed in the provided reference (<https://pubs.rsc.org/en/content/articlelanding/2014/cp/c4cp02749c>). This precedent supports

our interpretation that the ionomer coating modifies surface interactions, influencing the characterization of pore sizes.

We trust this response clarifies the subtle shifts observed in the pore size distributions of our catalysts upon ionomer coating.

Action taken:

Revision in Manuscript:

The reduction in mesoporosity can be attributed to the blocking of some mesopores that have microporous openings. This interpretation is supported by the fact that the lost volume of mesopores ($1.4 \text{ cm}^3/\text{g_catalyst}$) was substantially larger than the total volume of ionomers ($0.3 \text{ cm}^3/\text{g_catalyst}$), precluding the filling of mesopores by ionomers. Subtle variations in the pore width within the mesoporous region were also observed in both Pt/HOMC and Pt/Vulcan following the introduction of ionomer. Specifically, Pt/Vulcan experienced an increase in pore size from 2.0 nm to 2.5 nm, and Pt/HOMC from 4.0 nm to 4.8 nm. These changes are attributed to modifications in adsorbate-adsorbent interactions. For Pt/HOMC, the C-BET values for the uncoated and ionomer-coated catalysts changed from 21.03 to 3.98, respectively. The significant reduction in C-BET value with ionomer coating suggests weakened interaction dynamics between nitrogen and the catalyst surface, necessitating higher partial pressures to fill the same pores. This adjustment results in a rightward shift in the adsorption isotherms, and consequently, an apparent increase in the estimated pore sizes. These observations align with findings from studies using ionic liquids and silica⁴³. Despite these alterations, the mesopore openings in Pt/HOMC remain largely intact, indicating that the ionomer segments neither fill nor block the mesopore openings.

9. Ex-situ XAS comparison for the catalysts with and without ionomer addition should be provided to analyze potential discrepancies in ionomer distribution.

Response:

Thank you for your suggestion to include ex-situ XAS comparisons for the catalysts with and without ionomer addition. Unfortunately, due to constraints in acquiring additional synchrotron beamtime, we were only able to conduct XANES analysis for the three catalysts without ionomer. However, we have integrated these results with previously collected operando data from the catalysts in GDE with ionomer at open circuit potential (OCP). This combined analysis has been presented in Figure S16 of the Supplementary Information.

As depicted in Figure S16d, the influence of ionomer on the catalysts varies, following the order of Pt/Vulcan XC-72 > Pt/Ketjenblack > Pt/HOMC. This trend aligns with our model, suggesting that Pt/Vulcan XC-72 has the most intimate contact with the ionomer, significantly affecting its electrochemical environment. These findings offer insightful observations into the distribution and impact of ionomer across different catalyst supports.

Action taken:

Revision in SI:

Figure S16. XANES spectra at Pt L₃ edge for catalysts with and without ionomer addition: (a) Pt/HOMC, (b) Pt/Ketjenblack and (c) Pt/Vulcan XC-72. (d) $\Delta\mu$ XANES obtained by subtracting normalized absorption for catalyst without ionomer from that for catalyst with ionomer.

As depicted in Figure S16d, the influence of ionomer on the catalysts varies, following the order of Pt/Vulcan XC-72 > Pt/Ketjenblack > Pt/HOMC. This trend aligns with our model, suggesting that Pt/Vulcan XC-72 has the most intimate contact with the ionomer, significantly affecting its electrochemical environment. These findings offer insightful observations into the distribution and impact of ionomer across different catalyst supports.

10. A table illustrating the white line intensity change concerning potential variations for the three catalysts should be included for clearer comparison.

Response:

Thanks for the suggestion.

A table has been included in SI to aid the reading of changes in white line intensity.

Action taken:

Revision in SI:

Table S2. White line intensity change with potential variations.

Potential	Pt L ₃ edge white line intensity change ^a	Pt L ₂ edge white line intensity change ^a
-----------	---	---

vs RHE (V)	Pt/HOMC	Pt/Ketjenblack	Pt/Vulcan XC-72	Pt/HOMC	Pt/Ketjenblack	Pt/Vulcan XC-72
0.6	0.00244	0.00168	0.00640	-0.00128	-0.00490	0.00272
0.7	0.00641	0.00502	0.00869	0.00050	-0.00451	0.00056
0.8	0.00494	0.00527	0.01172	0.00098	-0.00324	-0.00234
0.9	0.01214	0.00825	0.01915	0.00562	-0.00133	0.00384
1.0	0.02238	0.02011	0.03615	0.01097	0.00622	0.01485

^a Values were calculated by subtracting white line intensity at 0.5 V.

REVIEWER COMMENTS

Reviewer #1 (Remarks to the Author):

From the "Response to the reviewers" and the revised manuscript, I can see that the authors have dealt with all my concerns well. So, now I believe it can be published.

Reviewer #2 (Remarks to the Author):

The authors have made several revisions to the work to support their interpretations.

1. The additional gas sorption measurements with CO₂@273K seem to confirm that the pore blocking seen in the N₂ isotherms are indeed due to Pt nanoparticles that bottleneck the porosity, but that CO₂ at room temperature overcomes the barrier. This is in line with the conceptual model being proposed in Figs 6 and 7, which is important. Their explanations for the gas sorption data resolve the incongruities in the initial manuscript.

2. It is really a shame the electrochemistry is not able to demonstrate a nice CV with visible Pt. I would recommend cycling past 1.05V, up to 1.23 or 1.25V to ensure the Pt is really being cleaned. But the explanation of the carbon capacitance being high relative to the Pt makes sense. The inability to evaluate the Pt surface with traditional electrochemistry is a big weakness to this approach with very dilute Pt, and hinders its comparison with literature. The use of the half cell is also unfortunate, since the authors claim it prevents them from using the CO displacement technique, due to multiple electrolytes being present. It is not obvious to me why an RDE approach, or a differential approach comparing multiple conditions cannot be used.

3. The additional details for the XANES are good, especially addressing the beam damage observed during preliminary measurements. Could the authors please add details for the approximate incident flux at the sample, the spot size of the beam on the sample, and the duration of the total exposure time, so that readers might determine how much integrated dose is being used here, relative to other work in the field.

One issue that was forgotten in the first round of review: this is not the first attempt at an "ionomer secluded" catalyst. Several different approaches have tried to achieve this, principally to reduce the poisoning effect of the ionomer on the Pt surface, but also for stability and other reasons. The best known is the NSTF from 3M, which was developed and optimized over more than a decade. Its ultimate failure, related to poor water management at high current density in full scale devices, is instructive here. The authors should probably engage with this long history and literature precedent, which guides the expected impact of the present manuscript, and all catalysts which use similar approaches.

Reviewer #3 (Remarks to the Author):

All concerns have been addressed. I suggest to publish as it is.

2nd Response to Reviewers for Nature Communications (NCOMMS-23-58195)

The comments by the reviewers are in *grey italic text* and the authors' replies are in **green text**, with revisions to the manuscript **highlighted in yellow**.

Reviewer #2 (Remarks to the Author):

The authors have made several revisions to the work to support their interpretations.

1. The additional gas sorption measurements with CO₂@273K seem to confirm that the pore blocking seen in the N₂ isotherms are indeed due to Pt nanoparticles that bottleneck the porosity, but that CO₂ at room temperature overcomes the barrier. This is in line with the conceptual model being proposed in Figs 6 and 7, which is important. Their explanations for the gas sorption data resolve the incongruities in the initial manuscript.

Response:

Thank you for acknowledging the revisions and additional data that strengthen the validation of our conceptual model. We are pleased that the new measurements and detailed explanations have addressed the previous concerns effectively.

2. It is really a shame the electrochemistry is not able to demonstrate a nice CV with visible Pt. I would recommend cycling past 1.05V, up to 1.23 or 1.25V to ensure the Pt is really being cleaned. But the explanation of the carbon capacitance being high relative to the Pt makes sense.

Response:

Thank you for acknowledging the impact of carbon capacitance on the absence of characteristic Pt peaks in our CV analyses. We appreciate your suggestion to extend the cycling potential range. While extending the cycling potential could – in principle - enhance Pt surface cleaning, we refrain from doing so due to potential carbon corrosion risks. Our catalysts are designed with uniform mesopores; higher potentials could jeopardize their structural integrity, which is central to our study's objectives. Literature indicates that excessive potential can lead to significant carbon degradation ([10.1016/j.jpowsour.2020.229434](https://doi.org/10.1016/j.jpowsour.2020.229434); [10.1149/2.0061806jes](https://doi.org/10.1149/2.0061806jes)). We take the view the absence of the Pt peaks is due primarily to the substantial relative capacitance of the carbon support, which we estimate to be as much as 64 times that of 40% Pt on carbon. This is reflected in the Pt ECSA to carbon surface area ratio, approximately 0.0015 for our Pt/Vulcan (estimated as $41\text{m}^2/\text{g}_{\text{Pt}} \times 0.78\% / 210\ \mu\text{C}\cdot\text{cm}^{-2}$) and 0.095 for HiSPEC4000 (estimated as $50\ \text{m}^2/\text{g}_{\text{Pt}} \times 40\% / 210\ \mu\text{C}\cdot\text{cm}^{-2}$).

3. *The inability to evaluate the Pt surface with traditional electrochemistry is a big weakness to this approach with very dilute Pt, and hinders its comparison with literature. The use of the half cell is also unfortunate, since the authors claim it prevents them from using the CO displacement technique, due to multiple electrolytes being present. It is not obvious to me why an RDE approach, or a differential approach comparing multiple conditions cannot be used.*

Response:

We acknowledge your concerns regarding our use of a very dilute Pt concentration and a gas diffusion half-cell. We chose the dilute Pt concentration strategically to maximize the utility of the mesopores of our carbon support and to specifically study the protective effects on Pt active sites, crucial for developing high-performance catalysts.

As it happens, we have done RDE measurements (see Fig S7): we did not observe significant difference between the activity of the different samples. This is because the three catalysts exhibit similar kinetic behaviour, and no notable differences were observed at the low current density in the RDE test, likely due to the limited solubility of O₂ in the electrolyte. It is also worth pointing that the limiting current density we observed in the RDE are more than 90% of its theoretical value (e.g., -6 mA/cm² for 1600 rpm in 0.1 M HClO₄), suggesting the catalysts and setup are properly cleaned despite the absence of characteristic Pt peaks. We did not perform CO displacement, due to safety restrictions in the specific laboratory where we conducted the RDE experiments, or CO displacement in an actual MEA cell; however, we agree that such experiments could be worthy of future studies.

Action taken:

Addition of discussion of CO displacement:

Future studies could benefit from CO displacement experiments to further quantify the interaction between ionomers and platinum surface, providing additional insights into catalyst behaviour under operational conditions.

3. *The additional details for the XANES are good, especially addressing the beam damage observed during preliminary measurements. Could the authors please add details for the approximate incident flux at the sample, the spot size of the beam on the sample, and the duration of the total exposure time, so that readers might determine how much integrated dose is being used here, relative to other work in the field.*

Response:

Thank you for your suggestion to provide more specific details on the XANES experiment setup. We have added the requested information to the Supplementary Information Table S3.

Action taken:

Addition of Supplementary Information Table S3:

Sample	Approx Beam Size (μm)	Approx Flux (ph/s)	Worst case time on Sample (HH:MM)
Pt/HOMC	100 x 100	4.3×10^{11}	08:35
Pt/Vulcan	100 x 100	4.3×10^{11}	07:53
Pt/ Ketjenblack	100 x 100	4.3×10^{11}	09:55

Table S3. Summary of Beam Conditions During XANES Measurements

This table details the experimental beam conditions for XANES spectroscopy conducted on various catalyst samples. The approximate beam size indicates the focused beam dimensions at the sample position. The flux values, sourced from Diamond Light Source B18 Si (111) during commissioning, have been adjusted based on ion chamber (I0) absorption to provide an approximation of the flux at the sample location. The 'Worst case time on sample' represents the maximum exposure duration for each sample, calculated based on the longest interval between the start times of sequential scans.

4. One issue that was forgotten in the first round of review: this is not the first attempt at an "ionomer secluded" catalyst. Several different approaches have tried to achieve this, principally to reduce the poisoning effect of the ionomer on the Pt surface, but also for stability and other reasons. The best known is the NSTF from 3M, which was developed and optimized over more than a decade. Its ultimate failure, related to poor water management at high current density in full scale devices, is instructive here. The authors should probably engage with this long history and literature precedent, which guides the expected impact of the present manuscript, and all catalysts which use similar approaches.

Response:

Thank you for highlighting the relevance of historical efforts in developing the NSTF from 3M. We have added a discussion on this topic in the Introduction section of our manuscript to acknowledge these precedents and draw lessons from their outcomes.

Action taken:

Revision in Manuscript highlighted in yellow:

However, the ionomer presence on the catalyst's surface, while improving proton transport and mechanical stability, can also have a poisoning effect²¹ and add an extra diffusion layer for O₂ transport.²² Nanostructured thin-film (NSTF) catalysts, developed by 3M, have been the most extensively studied ionomer-free electrodes in recent decades.²³ NSTF catalysts use highly oriented, crystalline organic whiskers as a support structure, onto which thin films of Pt or Pt alloys are sputter-coated.²⁴ This unique thin film structure enables direct contact between the catalyst and the electrolyte, thus eliminating the need for an ionomer to facilitate

proton transport. However, this thin structure also suffers from limited water-handling capacity.²⁵ The rate of water formation at the NSTF cathode is much higher than that of conventional Pt/C-ionomer electrodes, easily leading to flooding. Moreover, Pt/NSTF catalysts perform poorly under dry conditions due to reduced proton conductivity, making water management a critical challenge. Despite several material modification strategies aimed at addressing these issues,²⁶ the inherent difficulties of eliminating ionomer from the catalyst layer remain.

Various strategies have been employed to manipulate the ionomer's coverage on Pt surface including the ionomer's chemistry modification²⁸, molecular masking²⁹, regulating ionomer distribution by introducing N sites³⁰ and catalyst support engineering³¹.

Additional citations related to above discussion:

23 M. K. Debe, Nanostructured Thin Film Electrocatalysts for PEM Fuel Cells - A Tutorial on the Fundamental Characteristics and Practical Properties of NSTF Catalysts, *ECS Trans.*, 2012, **45**, 47–68.

24 M. K. Debe, R. T. Atanasoski and A. J. Steinbach, Nanostructured Thin Film Electrocatalysts - Current Status and Future Potential, *ECS Trans.*, 2011, **41**, 937.

25 P. K. Sinha, W. Gu, A. Kongkanand and E. Thompson, Performance of Nano Structured Thin Film (NSTF) Electrodes under Partially-Humidified Conditions, *J. Electrochem. Soc.*, 2011, **158**, B831.

26 A. Kongkanand, M. Dioguardi, C. Ji and E. L. Thompson, Improving Operational Robustness of NSTF Electrodes in PEM Fuel Cells, *J. Electrochem. Soc.*, 2012, **159**, F405–F411.